# The Secretion of Toxins and Other Exoproteins of *Cronobacter*: Role in Virulence, Adaption, and Persistence

**DOI:** 10.3390/microorganisms8020229

**Published:** 2020-02-08

**Authors:** Hyein Jang, Gopal R. Gopinath, Athmanya Eshwar, Shabarinath Srikumar, Scott Nguyen, Jayanthi Gangiredla, Isha R. Patel, Samantha B. Finkelstein, Flavia Negrete, JungHa Woo, YouYoung Lee, Séamus Fanning, Roger Stephan, Ben D. Tall, Angelika Lehner

**Affiliations:** 1Center for Food Safety and Applied Nutrition, U.S. Food and Drug Administration, Laurel, MD 20708, USA; hyein.jang@fda.hhs.gov (H.J.); Jayanthi.Gangiredla@fda.hhs.gov (J.G.); flavianegrete@yahoo.com (F.N.); junghaa12@gmail.com (J.W.); luy902@naver.com (Y.L.); 2Institute for Food Safety and Hygiene, University of Zurich, Zurich CH-8006 Zürich, Switzerland; athmanya.eshwar@uzh.ch (A.E.); stephanr@fsafety.uzh.ch (R.S.); lehnera@fsafety.uzh.ch (A.L.); 3UCD-Centre for Food Safety, Science Centre South, University College Dublin, Dublin Belfield, Dublin 4, D04 V1W8, Ireland; ssrikumar@uaeu.ac.ae (S.S.); scott.nguyen@ucd.ie (S.N.); sfanning@ucd.ie (S.F.)

**Keywords:** virulence factors, protein secretion systems, quorum sensing systems, outer membrane proteins, osmotic stress response, efflux pumps, plasmids, adherence factors, iron transport

## Abstract

*Cronobacter* species are considered an opportunistic group of foodborne pathogenic bacteria capable of causing both intestinal and systemic human disease. This review describes common virulence themes shared among the seven *Cronobacter* species and describes multiple exoproteins secreted by *Cronobacter*, many of which are bacterial toxins that may play a role in human disease. The review will particularly concentrate on the virulence factors secreted by *C. sakazakii*, *C. malonaticus,* and *C. turicensis*, which are the primary human pathogens of interest. It has been discovered that various species-specific virulence factors adversely affect a wide range of eukaryotic cell processes including protein synthesis, cell division, and ion secretion. Many of these factors are toxins which have been shown to also modulate the host immune response. These factors are encoded on a variety of mobile genetic elements such as plasmids and transposons; this genomic plasticity implies ongoing re-assortment of virulence factor genes which has complicated our efforts to categorize *Cronobacter* into sharply defined genomic pathotypes.

## 1. Introduction

*Cronobacter* species are considered an opportunistic group of foodborne pathogens capable of causing both intestinal and systemic human disease. There are seven species that are taxonomically recognized within the genus: *Cronobacter sakazakii, Cronobacter malonaticus, Cronobacter turicensis, Cronobacter muytjensii, Cronobacter dublinensis, Cronobacter universalis,* and *Cronobacter condimenti* [1,2,3]. Except for *C. condimenti*, all species of *Cronobacter* have been isolated from clinical specimens. *Cronobacter* has always been considered a neonatal pathogen, but it can infect other susceptible individuals such as older infants and elderly individuals alike and continues to attract media attention locally, nationally, and internationally [4,5,6,7,8,9,10,11,12,13,14,15]. Urmenyi and Franklin, in 1961, reported the first cases of fatal invasive newborn infections (meningitis) caused by *Cronobacter* species (reported as a yellow-pigmented *Enterobacter cloacae*) [16]. Furthermore, *Cronobacter* was elevated to a global foodborne and public health issue when contaminated lots of powdered infant formula (PIF) or follow-up formula (FUF) were epidemiologically linked to several neonatal and infant septicemia/meningitis outbreaks [17,18,19]. In addition to meningitis, the range of *Cronobacter* infantile infections have been extended now to include necrotizing enterocolitis (NEC) and bacteremia or sepsis; death can occur within hours from the onset of symptoms [4,5,6,7,8,9,11,13,16,17].

PIF was thought to be the source of neonatal/infantile infections. However, it is clear now that contamination of reconstituted PIF can occur intrinsically and extrinsically, although the main reservoir(s) and routes(s) of contamination have yet to be determined [18,19,20]. Jason reported surveillance data on 82 *Cronobacter* cases (between 1958 and 2010) and showed that these infants became ill (defined here as a confirmed culture-positive case of septicemia or meningitis) after ingesting breast milk exclusively (without consumption of PIF, FUF, or powdered human milk fortifiers) prior to illness onset [11]. Friedemann had also reported similar observations [20]. To underscore this point, Bowen et al. [21] and McMullan [22] recently reported infantile cases of *C. sakazakii* septicemia/meningitis where these infants only consumed expressed maternal milk (EMM) during the first weeks after birth. Contaminated personal breast pumps were found to be the source of the contamination. Pulsed-field gel electrophoresis (PFGE) and whole genome sequencing (WGS) analyses of isolates determined that the clinical isolates were indistinguishable with those cultured from a contaminated breast pump and a home kitchen sink drain in the first case and the breast pump in the latter case. Together, these data suggest that breast feeding and EMM recommendations from health care individuals need to be better communicated to nursing mothers. Of equal significance is that *Cronobacter* species are largely more ecologically widespread and have been found associated with many types of foods besides infant formula products. For example, *Cronobacter* species have been found associated with dried dairy protein products (milk and cheese protein powders), cereals, candies such as licorice and lemon-flavored cough drops, dried spices, teas, nuts, herbs, and pastas and water [4,23,24,25]. It has also been found associated with many different ready-to-eat and frozen vegetables, insect body surfaces and intestinal contents, and man-made environments such as PIF or dairy powder production facilities, and household sink drains [23,24,26,27,28,29].

*Cronobacter* display a variety of unique features which support the organism’s capability to survive under various stressful growth environments and these attributes may also be beneficial to the organism when it interacts with humans [27,30]. This review will discuss common themes of virulence shared among the seven *Cronobacter* species and describe multiple exoproteins secreted by *Cronobacter,* many of which are bacterial toxins which may play a role in human disease. Even though all but *C. condimenti* has been associated with human disease we will particularly concentrate on some proteins or virulence factors secreted by *C. sakazakii*, *C. malonaticus* and *C. turicensis*, which are the primary human pathogens of interest. 

## 2. Common Themes in *Cronobacter* Virulence

Like with most enteric pathogens that interact with humans, the human mucosa or mucus membrane is the first site of contact that allows *Cronobacter* species to follow a well-recognized bacterial infection stratagem comprising of: (i) colonization at a mucosal site, e.g., intestinal, respiratory, or the urinary tract epithelia, (ii) circumvention, subversion, and exploitation of host defenses, e.g., invasion of intestinal epithelial cells or internalization and survival within phagocytic cells, which may also provide the pathogen a niche with less competition from other microorganisms, as well as provision of new and rich nutrients, (iii) systemic spread and multiplication, e.g., within the blood, phagocytes, or at extra-intestinal sites such as the meninges, and (iv) host damage, e.g., through expression of exoproteins such as toxins and/or damage due to pro-inflammatory modulation of the host immune system. In fact, Cruz-Córdova et al. provided evidence that the flagellum of *Cronobacter* species can induce inflammatory cytokines, such as IL-8, TNF-α, and IL-10 [31]. 

One of the most highly conserved phenotype expressed by *Cronobacter* species is their capacity to colonize the intestinal mucosal surface and compete for nutrients in the presence of the gut microbiome, despite the action of peristalsis [32]. The expression of adherence factors such as fimbriae or pili and outer membrane protein adhesins is a common trait possessed by all *Cronobacter* strains and represents various classes of secreted and polymerized exoproteins [32,33]. However, other than genomic findings showing the presence of such genes and gene clusters within respective genomes, little information is known about mechanisms of intestinal adherence. On the other hand, biofilm formation by *Cronobacter* species on common materials used in PIF manufacturing (and other food production environments) has been well documented both experimentally and within commercial manufacturing environments [27,34,35,36,37]. A biofilm is comprised of microorganisms and occurs when cells stick to each other as they colonize a surface. These adherent cells become embedded within a matrix of extracellular polymeric substances. Additionally, the flagellum subunit FliC found to be involved in the *Cronobacter* auto agglutination phenotype was noted by Hoeflinger and Miller [38]. However, the role of flagella in biofilm formation needs further investigation. The formation of biofilms is mediated by quorum sensing and is based on the synthesis, secretion, and cellular detection of signaling molecules [39]. Extracellular concentrations of these signaling molecules are sensed by cells of the pathogen and upon reaching a population density-dependent threshold, allows the induction of targeted gene expression of the entire cell population in a highly coordinated and controlled fashion. 

Once host colonization has been established, the pathogenomic strategies that *Cronobacter* strains possess can be remarkably different. General disease paradigms that have been described for infant *Cronobacter* infections include: necrotizing enterocolitis, pneumonia, septicemia or meningitis. Additionally, similar infections are observed in adults such as septicemia and pneumonia; but other types of adult infections have also emerged such as wound and urinary tract infections (UTI) [10,13,14,15]. 

As is the case for many enteric pathogens, another common feature possessed among *Cronobacter* species is that they carry plasmids, which are known to be involved in contributing to genomic plasticity, bacterial virulence, and survival [40]. In general, such plasmids have been found among members of the *Enterobacteriaceae* that encode a diversity of virulence factors, such as harboring genes for antibiotic resistance, toxins, adherence factors, and secretion systems (types 3, 4, and 6), and it is thought that plasmid-borne virulence genes (or gene clusters) are acquired through horizontal gene transfer (HGT) [40,41,42,43,44,45,46]. In silico analysis of such plasmids harbored by *Cronobacter* species suggests that this common theme holds true here as well [47]. In fact, Muytjens et al. was the first group to identify and characterize plasmids possessed by clinical *Cronobacter* strains (reported as *E. sakazakii*) [48]. The strains were acquired during a 6-year surveillance study of neonatal meningitis and septicemia cases. Other researchers performed similar studies [49,50,51], which also revealed that such strains (reported as *E. sakazakii*) possessed multiple sized plasmids, and these strains were isolated from ill infants, utensils used to prepare infant formula which was consumed by the infants, and from containers of PIF formulations. Since then, whole genome sequencing (WGS) has allowed for the study of plasmids at the genomic level and during 2010–2011, the first closed genomes of *C. sakazakii* strain BAA-894 and *C. turicensis* strain LMG23827^T^ were made available [33,52]. Franco et al. [47] used this information to characterize two plasmids, pESA3 and pCTU1 harbored by these strains. WGS analysis revealed that plasmids pESA3 (131 kbp), pCUNV1 (129 kbp), and pCTU1 (138 kbp), respectively encode a single and shared RepFIB-like (incompatibility class) origin of replication gene, *repA*, as well as two iron acquisition gene clusters, *eitCBAD* (ABC ferric-iron transporter) and *iucABCD/iutA* (hydroxamate-type siderophore aerobactin, named Cronobactin) [33,52,53]. Figure 1 shows the sequence homology shared between these three virulence plasmids. The *iucABCD/iutA* gene cluster is the only known *Cronobacter* siderophore present among the seven species even though multiple iron acquisition systems have been described [54]. This redundancy in iron acquisition genes (gene clusters) may reflect the niche diversity occupied by *Cronobacter* species (such as eukaryotic plants, PIF manufacturing facilities, and flies to name a few), as well as the disposition and bioavailability of various sources of iron within these unique environments.

Reports by Liu et al. [57] and Zogaj et al. [58] showed that *Cronobacter* can colonize the gastrointestinal tract without causing disease. However, more epidemiological information is required to understand its function as a member of the gut microbiota, and whether its presence represents transient colonization of particular strains or other types of host-associations and how different bacteria adapt and evolve for example, many bacteria are “commensal pathogens”, i.e., able to cause disease in some people and be harmless in others.

## 3. *Cronobacter sakazakii*

*Cronobacter sakazakii*, as described by Iversen et al. [1] consists of a group of strains previously reported by Farmer et al. [59] as belonging to biogroups 1–4, 7, 8, 11, and 13 of the former *E. sakazakii* species epithet. *C. sakazakii* is the predominant *Cronobacter* species associated with infantile and adult illnesses [4,5,6,7,8,13,14,15,18]. Using the multi-locus sequence typing (MLST) scheme described by Baldwin et al. [60], Joseph et al. [61], and Joseph and Forsythe [62] showed an association among particular *C. sakazakii* sequence types (ST), which are found with specific types of illnesses [63,64]. For example, *C. sakazakii* ST4 was found to be associated with infantile septicemia and meningitis cases [15,64]. Other clinically important STs include *C. sakazakii* ST1, ST8, ST12, ST15, ST40, ST107, ST110 and ST111. Interestingly, Joseph and Forsythe reported that *C. sakazakii* possessing the ST profiles of ST15, ST97, ST107, ST108, and ST110 are highly related to those possessing the ST4 allelic profile and represent a group of strains forming a clonal complex, CC4 [62]. Clonal complexes represent strains that have single, double, or triple loci variants among the seven MLST alleles and just one nucleotide difference in one locus results in an assignment of a different allelic profile number. Furthermore, *C. sakazakii* possessing the ST12 alleles have been found associated with necrotizing enterocolitis cases [65]. *C. sakazakii* is strongly associated with severe and often fatal cases of necrotizing enterocolitis and meningitis in neonates and infants. 

Interestingly, whole genome sequencing analysis has revealed that *C. sakazakii* possess a *nanAKT* gene cluster which allows for the utilization of exogenous sialic acid which is found in breast milk, infant formula, intestinal mucin, and gangliosides in the brain. Currently only a few strains have been analyzed, but laboratory studies reported by Joseph et al. [66] confirmed that only *C. sakazakii*, and not the other six *Cronobacter* species, was able import and catabolize sialic acid which also suggests that the organism may have adapted to the human host driven by these known reserves of sialic acid. 

A study reported by Alsonosi et al. identified 51 isolates from two hospitals located in the Czech Republic and they found a preponderance of *C. sakazakii* ST4 strains among adult cases of pneumonia, wound infections, and UTIs, which suggests that *C. sakazakii* ST4 may also be emerging as an adult pathogen [15]. Currently there is not enough epidemiological information to tease apart community-acquired infections from nosocomial infections. In addition to the clinically significant ST1, ST4, and ST8 strains being found associated with PIF manufacturing environments in North America, Europe and China, other important STs are ST31, ST40, ST64, ST83, ST103, ST196, ST194, and ST190 [26,67].

## 4. *Cronobacter malonaticus*

*Cronobacter malonaticus*, as described by Iversen et al. [1], comprises strains that were previously reported by Farmer et al. [59] as belonging to biogroups 5, 9, and 14 of the former *E. sakazakii* species epithet. As the species’ name implies, *C. malonaticus* strains can utilize malonate. Malonate utilization is an important differential trait and well recognized as being possessed by six of the seven *Cronobacter* species. Interestingly, Alsonosi et al. found that 33% of the 51 *Cronobacter* hospital-related cases (17/51, irrespective of age) were caused by *C. malonaticus* and came from sputum samples in 13 of the 17 cases [15]. These data suggest that *C. malonaticus* may have a greater epidemiological significance with respiratory infections than what was previously thought. The sequence type for these strains was determined to be ST7. Querying the *Cronobacter* MLST website (Available online: https://pubmlst.org/cronobacter/, last accessed 12/20/2019) for which *C. malonaticus* STs were involved in clinical cases showed a predominance of ST7 strains (34/69, 49%) or related strains that were associated with CC7 (41/69, 59%) followed by strains identified as ST11, ST60, ST307, and ST84 to name a few of the other noted STs [60]. The fact that 12 of these 17 cases (70%) reported by these authors involved infections associated with individuals of 5 years or older, also supports the fact that *C. malonaticus* may be emerging as a pathogen of older children and the elderly [15,65]. Interestingly, Iversen et al. [1], reported that malonate utilization was present in approximately 5% of *C. sakazakii* strains. Subsequently, using in parallel a novel DNA microarray and whole genome sequencing analyses to identify *Cronobacter* species Gopinath et al. described a nine gene malonate operon (~7.7 kbp in size) that was located between two highly conserved flanking genes, *gyrB* and *katG* in ST64 *C. sakazakii* strains obtained from clinical sources, foods and food production facilities in Europe, southern Asia, China, and USA [68]. Of note, the presence of *gyrB* and *katG* was also found to be conserved among all *Cronobacter* species even malonate-negative *C. sakazakii* strains of other STs; however, instead of the malonate utilization gene cluster, there is a 323–325 bp nucleotide region [68]. The under appreciation of malonate-positive *C. sakazakii* strains that are associated with foods presents an epidemiological problem if phenotypic identification schemes alone are used in species identity. However, to date, it is unknown whether other *C. sakazakii* STs possess a malonate utilization operon or if this genotype is exclusively found in *C. sakazakii* ST64 strains. 

Based on WGS studies by Ogrodzki and Forsythe, an important capsular typing scheme using the K-antigen and colanic acid (CA) biosynthesis regions was described [69]. It is based on information coming from analyzing 104 *Cronobacter* strains for the presence of a previously uncharacterized *Cronobacter* capsular region (*kps*) [69]. The region was like the well-described K-antigen gene cluster of *Escherichia coli*. The gene cluster is comprised of three regions: K-antigen region 1 (*kpsEDCS*) and region 3 (*kpsTM*) are conserved across the genus, and there are two variants of region 2 that was found. Genes associated with K-antigen type 1 are present in all seven species of *Cronobacter*. However, the prevalence and distribution of K-antigen type 2 (specifically of interest is the capsular profile K-antigen gene cluster 2–colanic acid gene cluster 2– cellulose positive profile) are not as prevalent in all seven species for this antigenic profile is found only in *C. sakazakii*, *C. malonaticus*, *C. turicensis*, and *C. dublinensis*. It is thought that the presence of this capsular type specifically in *C. sakazakii* and *C. malonaticus* strains may confer a favorable phenotype important in desiccation resistance, persistence, and serum resistance as well as increased macrophage survival, resulting in a more physiologically fit pathogen [69].

## 5. *Cronobacter turicensis*

*Cronobacter turicensis*, as described by Iversen et al. [1], comprises strains which were previously reported by Farmer et al. [59] as belonging to biogroup 16 of the former *E. sakazakii* species epithet [1,70]. Much less is known about this species, the first clinical case of *C. turicensis* infection was reported in 2005 by Stephan et al. [52] as a strain that caused the death of two neonates in 2005. Querying the *Cronobacter* MLST website (Available online: https://pubmlst.org/cronobacter, last accessed 12/20/2019) for C. turicensis STs involved in clinical cases showed that there were only 10 strains which came from clinical cases and these cases involved strains possessing the following STs: ST5, ST19, ST309, ST350, and ST636 determinants and seven of these ten strains came from Europe [60]. However, there were 100 *C. turicensis* records registered in the *Cronobacter* MLST site which were obtained from a variety of sources including milk powder, water, spices, tea, vegetables, insects, manufacturing environments, and ready to eat foods, with multiple ST designations. 

## 6. *Cronobacter* Secreted Toxins/Exoproteins

### 6.1. Cronobacter Plasminogen Activator

Whole genome sequencing studies of *C. sakazakii* BAA-894 demonstrated the presence of plasmid pESA3 [33] and in silico analysis showed that it encodes an outer membrane protease with significant amino acid sequence homology to proteins belonging to the omptin family [47,71]. Omptins are outer membrane proteins which are expressed by several members of the *Enterobacteriaceae* [72,73]. Omptins are known bacterial virulence factors that can function as proteases, adhesins, or invasins [72,73,74]. Franco et al. [47] showed that this plasmid-borne OmptinT-like protease now named *Cronobacter* plasminogen activator (Cpa) had significant identity to proteases that belong to the Pla subfamily of omptins such as PgtE which is expressed by *Salmonella enterica* [71]. Other omptin-like proteases include Pla of *Yersinia pestis*, and PlaA of *Erwinia* spp. Figure 2 shows the phylogenetic relatedness among the OmptinT family of proteins including the phylogenetic relatedness of Cpa possessed by *C. sakazakii* and *C. universalis* which is also mapped to plasmid pCUNV1. Furthermore, Franco et al. summarized the proteolytic activity of members of this subfamily of proteases and showed that Cpa expressed by *C. sakazakii* degraded several host serum proteins, including circulating complement components [71]. It is conjectured that the degradation of these complement components by Cpa will allow systemically circulating *Cronobacter* cells to be protected from complement-dependent serum killing. Moreover, Cpa like other Pla-like proteins is thought to cause unrestrained plasmin activation by transforming plasminogen to plasmin, and inactivation of plasmin inhibitor α2-antiplasmin (α2-AP). Together, these findings suggest that Cpa expressed by *C. sakazakii* could proteolytically cleave complement components C3, C3a, and C4b and cause rapid activation of plasminogen and inactivation of α2-AP. Figure 3 shows schematically how Cpa may interact with plasminogen and α2-AP causing plasminogen to be continuously converted to plasmin which then allows for uncontrolled degradation of fibrin clots and extracellular matrix proteins which will further promote systemic spread of the pathogen. These results suggest that Cpa is an important virulence factor involved in serum resistance, as well as in the systemic spread of *C. sakazakii*. Unlike Pla (expressed by *Y. pestis*), it is not known whether plasmin binds to the *Cronobacter* bacterial cell surface, but it is known that similar plasma proteins such as fibronectin binds to the *Cronobacter* bacterial surface [75]. The presence of *cpa* (pESA3p05434) encoded on pESA3-like plasmids was thought to have evolved from a prototypical plasmid backbone through the co-integration or deletion of virulence determinants in each of the *Cronobacter* species. The *cpa* gene in *C. sakazakii* strain BAA-894 was found to be flanked upstream by an MFS transporter homologue and downstream by *cpmJK*, encoding proteins potentially involved in carbapenem resistance [47]. Interestingly, the *cpa*-flanking regions on plasmid pESA3 are maintained on plasmid pCTU1, the virulence plasmid harbored by *C. turicensis*; however, instead of the 1427-bp nucleotide region containing *cpa*, plasmid pCTU1 has a unique 37-bp region [47]. Furthermore, noted regulatory features that were found associated with this genomic region are palindromic inverted repeats (10–13 nucleotides in size which are separated by a 10-bp spacer). Furthermore, Franco et al. showed that the conserved plasmid pCTU1 region is located upstream of this inverted repeat, while the *cpa* locus on plasmid pESA3 is located downstream [47]. These authors hypothesized that the inverted repeat is a transposon attachment site, which would explain the acquisition, evolution, and presence of *cpa* on pESA3 or its absence on pCTU1. Taken together, these results suggest that these virulence plasmids have undergone microevolution or the co-integration or deletion of plasmid genetic attributes, which potentially may continue leading to the acquisition (or deletion) of other virulence genes. Furthermore, PCR analysis using primers designed to detect *cpa* showed that most (173/177, 98%) *C. sakazakii* do harbor *cpa* [47]. In addition, two *C. universalis* strains were also PCR-positive for *cpa*. Strains of the other *Cronobacter* species were PCR-negative for *cpa*, providing evolutionary evidence that the *cpa* locus may be a species-specific locus for *C. sakazakii* and *C. universalis*. Furthermore, Eshwar et al. examined a *C. sakazakii* mutant (ATCC BAA-894*Δcpa*) deficient in *cpa* and compared its virulence with the wild type strain BAA-894 and complemented strain ATCC BAA-894*Δcpa*/pQE30::*cpa* (with *cpa* in trans) in the Zebrafish infection model [76]. The *cpa*-deficient mutant (*C. sakazakii* ATCC BAA-894*Δcpa*) exhibited a 10% mortality rate compared to an 80% mortality rate with the wild type parental strain (ATCC BAA-894). Additionally, virulence was partially restored (40% mortality rate) with the complemented strain. Interestingly, growth of these strains in the zebrafish embryo model over time showed that until 24 h post infection (hpi), the growth slowed in the mutant followed by a sharp drop in bacterial counts of the mutant at 48 hpi, suggesting that the embryos could eliminate the *cpa* mutant strain by that time point [76]. Of note, ST8 *C. sakazakii* strains which possess the pESA3 virulence plasmid such as the clinical *C. sakazakii* species type strain ATCC 29544^T^ do not possess *cpa,* yet this strain is extremely virulent. These results suggest that other factors besides *cpa* are responsible for illness. 

### 6.2. Zinc Metalloprotease

Many pathogens produce bacterial metalloproteases which require basal metal ions such as zinc for catalytic activity and are thought to play a role in virulence in a variety of animal and plant hosts [78]. For example, the metalloprotease expressed by *Vibrio vulnificus* (designated VVP) causes serious hemorrhagic skin destruction through the breakdown of vascular extracellular matrices (e.g., endothelial basement membranes), and are especially active towards basal lamina-based type IV collagen fibers that form the scaffold and structural membrane support of cells [79]. *V. vulnificus* VVP also causes fluid accumulation or edema in tissues through induction of exocytotic histamine release from mast cells and/or activation of the factor XII-plasma kallikrein–kinin cascade [80]. These metalloproteases have both enzymatic and host immune modulation activities. Metalloproteases can also disassociate iron from heme that is complexed with specific host serum protein carriers [81]. They can degrade other plasma proteins and target tissue membrane proteins involved in cellular invasion, meningitis, and periodontal disease [81]. It is also thought that metalloproteases cause destruction of endothelial cell membranes which are associated with capillary vessels which lead to leakage of blood components into surrounding tissues, thereby enhancing the dissemination of bacteria from entry sites of infection into systemic circulation and being finally translocated to a target tissue site such as the blood-brain barrier (BBB) [81,82]. Kothary et al. described a zinc-containing metalloprotease (Zpx) expressed by *Cronobacter* species and showed that Zpx caused rounding (and eventual cell membrane damage) of Chinese Hamster ovary (CHO) cells [83]. Results showed that the proteolytic activity was cell surface-associated and that the metalloprotease is not secreted well. Even though these authors found that Zpx shares significant homology with other bacterial metalloproteases; an indispensable motif which is required for secretion of proteins was not found for Zpx. Nonetheless, these results support the finding that the protease may be cell bound. Interestingly, the observation that the metalloprotease may be cell bound suggests that an intimate bacteria-to-host cell contact may be required for initiation of its cytopathic effects through degradation of soluble proteins [83]. Further studies by this group showed that *zpx* was present in all *Cronobacter* species examined and that it possessed collagenolytic activity, but not elastinolytic activity [76,84]. The role of this protease in disease may be its involvement in necrosis and cellular damage in neonates with necrotizing enterocolitis; it may also be responsible for the pathology seen in meningitis [85]. Using the zebrafish embryo model, Eshwar et al. showed that the virulence of a *C. turicensis* LMG 23827^T^*Δzpx* mutant was diminished by 60% and virulence was restored to a large extent (80%) in experiments using a complemented mutant [84]. In another study, Eshwar et al. showed that the matrix metallopeptidase 9 (MMP-9), a eukaryotic proteinase which cleaves extracellular proteins such as collagen, was the substrate for Zpx and that this metallo-enzyme induces the expression of MMP-9 and led to a yet-undescribed mutual cross-talk between two proteases of bacterial and a host origins [84]. 

### 6.3. Hemolysins

A hemolysin (*hly*) gene was identified as a hemolysin III homolog (COG1272) by Cruz et al. [86]. Since then, several investigators have predicted that all *Cronobacter* species may possess a hemolysin III homolog (COG1272) [30,49,80]. However, strains identified to the species level (reported as *E. sakazakii*) by using 16S rRNA gene sequences and PCR designed to target the hemolysin III homolog gene showed that some strains possessed the gene, while others did not [86]. Singh et al. characterized the beta-hemolytic activity of several *C. sakazakii* strains isolated from food, soil, and water; these strains were PCR-positive for the COG1272 gene [87]. To better appreciate the gene prevalence, distribution, and phylogenetic relatedness of hemolysin III homolog COG1272 gene, Jang et al. [88] performed PCR, microarray and WGS analyses on over 300 *Cronobacter* strains of all seven species with their identities confirmed using both the *rpoB* and *cgcA* species-specific PCR assays as described by Stoop et al. [89], Lehner et al. [90] and Carter et al. [91], and showed that they do possess a hemolysin III COG1272 gene homolog, but the PCR primers described by Cruz et al. [86] may not detect all COG1272 orthologues in every *Cronobacter* species uniformly. Additionally, three other hemolysin genes were described by using this parallel next generation DNA-based approach and include alleles for a cystathionine beta synthase (CBS) domain containing hemolysin, a putative hemolysin, and a 21-kDa hemolysin [88]. Furthermore, Umeda et al. recently reported that 57 *Cronobacter* strains showed β-hemolytic activity against guinea pig, horse, and rabbit erythrocytes [92]. However, using sheep erythrocytes, the majority of strains (53/57; 92.9%) exhibited α-hemolysis activity. Taken together, more in-depth genetic studies are needed to assign functionality of these various hemolysin genes to the corresponding phenotype.

### 6.4. Enterotoxin

The suckling mouse assay, as described by Richardson [93], is used to study diarrhea caused by enterotoxin activity of many enteric pathogens. To determine if *Cronobacter* species (identified as *E. sakazakii*) could cause fluid accumulation (a measure of enterotoxin production) in suckling mice, Pagotto et al. tested 18 isolates [85]. Four of the eighteen strains caused fluid accumulation, suggesting that these strains may produce an enterotoxin [85]. Raghav and Aggarwal purified a 66-kDa secreted exoprotein using ammonium sulfate precipitation, followed by DEAE-cellulose ion exchange and desalting with SephadexTM G-100 [94]. The authors then used the suckling mouse assay to follow the isolation of the enterotoxin in each purification step. Toxin activity was undetectable in toxin preparations heated at 100°C and held for 30 min but was somewhat stable at 90°C held for 30 min [94]. To date, no gene has been assigned to this protein.

### 6.5. Macrophage Infectivity Potentiator (FkpA) 

Internalization of invading pathogens by macrophages is an innate immune process and formulates a host’s primary defense toward eliminating invasive pathogens. The capability of some pathogens to survive, persist, and multiply within macrophages is critical for their systemic survival, and as a primary step in development of severe illnesses such as sepsis or meningitis [32]. Studies by Horne et al. [95] and Humphreys et al. [96] suggest that the survival and persistence of *Salmonella* in murine and human macrophage cells may be affected by periplasmically located *cis-trans* prolyl isomerases (PPIases) such as Fkp. *Cis-trans* prolyl isomerases may be related to the *mip* (macrophage infectivity potentiator) gene [97]. Eshwar et al. queried GenBank for homologies of this gene in available *Cronobacter* genomes and found the presence of *fkpA*-like gene homologs in these organisms [97]. These authors evaluated the intracellular survival of FkpA variants in human macrophages by knocking out the *fkpA* genes in *C. turicensis* and *C. universalis*. Even though macrophage survival and replication varied among *Cronobacter* strains due to the presence of amino acid sequence variations in the respective FkpA proteins, their results provided convincing evidence that FkpA must be considered a virulence factor. These results also show that FkpA is expressed and possibly released intracellularly within the macrophage [97].

## 7. Bacterial Protein Secretion Systems in *Cronobacter*

Through the span of evolutionary time, bacteria have developed highly specialized systems to transport and secrete small molecules, proteins, and DNA [98]. The secretion of these substrates play key roles in how bacteria respond to their environment and in several important host-associated biological activities such as adaption, adherence, pathogenicity, persistence, and survival. The eventual outcome of the secretion process allows intracellular substrates to either be released extracellularly into the environment, remain cell-surface associated, or they are secreted through specialized outwardly directed flagellum-like structures (called injectosomes) into an adjacent cell (of either eukaryotic or bacterial origins). Five secretion systems that function to passage proteins across the cell membrane and the Gram-negative cell wall/outer membrane (OM) have been described and are classified as follows: type 1 secretion system (T1SS), T2SS, T3SS, T4SS, and T6SS. Another secretion system called T5SS involves secretion though just the OM and includes assembly systems associated with type 1 fimbriae and curli. The secretion process triggers what seems to be a strictly regulated response and aids in the recognition by the bacterial cell of the presence of host receptors or other host proteins. As an overview, secretion systems described in this review will concentrate on those protein secretion systems carried by *Cronobacter* species and will illustrate their role in making a more physiologically fit pathogen through the secretion of toxins and other proteins. To date, *Cronobacter* species have been found to possess T1SS, T2SS, T4SS, T5SS, T6SS, but no T3SS have been found.

### 7.1. Type 1 and 2 Secretion Systems

In general, T1SSs of Gram-negative bacteria allow for the secretion of a variety of substrates which are directly delivered from the bacterial cytoplasm into the extracellular milieu. Examples of such proteins are HasA (amino acid Heme-binding protein A; pfam06438) a heme acquisition protein or hemophore expressed by *Yersinia*, *Serratia*, and *Pseudomonas* species and many hemolysins such as *E. coli* α-hemolysin [98]. Due to their overall protein ultrastructure and homologies associated with multiple protein components, T1SSs are also closely related to the resistance-nodulation-division (RND) family of multidrug efflux pumps [92]. From structural and functional data, a simple secretion process for the T1SS and RND efflux pumps has been proposed [98]. Both systems form a tripartite double-membrane-spanning channel with an ATP-binding cassette transporter (ABC transporter) family protein which is also called an inner membrane component (IMC). Additionally, other proteins associated with the T1SS complex include a periplasmic adaptor protein (referred to as the membrane fusion protein) and TolC (an OM protein channel) common among Gram-negative bacteria [98]. The IMC component is involved in substrate recognition by identifying a glycine-rich motif (Gly-Gly-X-Gly-X-Asp) that is usually present as a repeat in the carboxyl terminus of its substrates, but as mentioned previously this motif is not found in the Zpx metalloprotease protein [83]. Presently functional studies of T1SS genes and their role in the secretion of toxins in *Cronobacter* are lacking. Another difference between T1SS and RND efflux pumps is that T1SS secretion is driven in an ATP-dependent manner, whereas secretion by RND efflux pumps uses a proton gradient connected to the secretion/uptake of respective substrates [98]. 

Type 2 secretion systems are also found in a wide variety of pathogenic and non-pathogenic bacteria. Several major differences exist between T2SSs and T1SS; T2SSs secrete folded proteins from the periplasm into the extracellular environment and are usually more complex than T1SS in that they are composed of 12–15 membrane spanning protein components [98]. Examples of T2SS proteins in other pathogens include hydrolyzing enzymes, such as pseudolysin of *Pseudomonas aeruginosa*, pullulanase of *Klebsiella pneumoniae* (that are important for host bacterial survival and growth within an environmental niche), and toxins (e.g., cholera toxin of *V. cholerae*) [98]. Two such *Cronobacter* proteins, thought to be secreted by a T2SS pathway, are the enzymes α-glucosidase and β-cellobiosidase. These two enzymes are used in differentiating *Cronobacter* species from other *Enterobacteriaceae* family members and play a role in carbohydrate metabolism [1]. Another major difference between T2SS and T1SSs is that T2SSs contain a pseudopilus which is in contrast to other pili used in adherence; the T2SS pseudopilus remains confined within the bacterial cell’s T2SS periplasmic secretome [92]. Similarly, type IV pili are evolutionarily related to the T2SS, and they share many similar structural and functional features [98].

### 7.2. Type 4 Secretion System

Type 4 secretion systems have the distinct ability among the various secretion systems to mediate the translocation of both DNA and proteins into bacterial and eukaryotic target cells through direct contact with a recipient cell. T4SSs are found in Gram-negative and Gram-positive bacteria and in some *Archaea* species [98]. Three categories of T4SS have been described; (i) conjugation systems which transfers DNA to recipient cells from donor cells by a contact-dependent process, (ii) T4SSs involved in pathogenicity by delivery of effector molecules or toxic proteins into eukaryotic host cells, and (iii) T4SSs involved in transfer of DNA to/or from the extracellular milieu [99]. Plasmid conjugation is mediated through a specific pilus structure. It is thought that certain bacteriophage uses T4SS pili as receptors [100]. Conjugation in bacteria is very common and the gene cluster responsible for T4SS secretion and pilus assembly are located on plasmids such as pESA2, SP291-2, and pCTU2 in *Cronobacter* [27,33,47,52,53]. However, there are some isolated genes located on the chromosome, but these are not in operonic form.

A second set of conjugative components include the “integrative and conjugative elements” (ICEs) That many bacteria possess. These are first excised from the chromosome of the donor cell, and then translocated to the recipient cell after a circular intermediate is formed. Once translocated, the ICE then reintegrates back into that cell’s chromosome [101]. Grim et al. reported that both *C. muytjensii* and *C. universalis* possess an ICE; the ICE in *C. muytjensii* is found in Genomic region 27 [102]. In some bacterial pathogens such as *Helicobacter pylori* and *Bordetella pertussis*, T4SSs also deliver effector proteins into the cytoplasm of a host cell to support bacterial intracellular survival [98]. Some examples of T4SS gene clusters found in other enteric species include the Ti plasmid of *Agrobacterium tumefaciens* and the conjugative plasmid pKM101 and plasmid R388 of *E. coli* [98]. Examples of proteins secreted by T4SSs include the pertussis toxin and CagA of *H. pylori* [98]. The gene cluster for the *Cronobacter* T4SS consists of genes encoding 12 proteins (VirB1-B11 and VirD4) and is found on a plasmid pESA2/pCTU2 [33,47,52]. A survey for prevalence and distribution of the T4SS pEAS2-like plasmids (synonymous with pCTU2 and pSP291-2) among 570 *Cronobacter* strains representing the seven species is shown in Table 1. The T4SS containing plasmid is found in approximately 4% of the strains. It is interesting to speculate on the evolutionary significance of *Cronobacter* strains harboring a plasmid with T4SS loci in that it adds to other lines of evidence suggesting that the environmental origins and ancestral econiche for *Cronobacter* species may be with eukaryotic plants [103,104]. Presently, it remains unanswered why most *Cronobacter* species have lost this plasmid (Table 1).

### 7.3. Type 5 Secretion System

The Type 5 secretion system or “autotransporter secretion pathway” is a unique secretion mechanism, in which an autotransporter C-terminal domain forms a pore for the secretion of the N terminal domain of the protein through the cell wall/outer membrane [105]. The substrate first must enter the periplasmic space usually through the activity of a SecYEG translocon [98]. As described by Henderson et al. autotransporters can facilitate several virulence mechanisms, such as the expression of adhesins used to colonize host cells, and actin-promoted bacterial intracellular mobility [106]. Proteins secreted by T5SS pathways need to possess three important domains; (i) a N-terminal targeting motif (amino-terminal leader sequence) that functions as a signal peptide to mediate translocation across the inner membrane using a SecYEG translocon, (ii) a carboxy-terminal translocation domain that forms a beta-barrel pore allowing the protein to be secreted through the OM, and (iii) the secreted mature protein [106,107]. Recently, Kothary et al. showed that an autotransporter protein (< 100kDa in size) was captured as a protein component within outer membrane vesicles (OMVs) of *C. sakazakii, C. malonaticus,* and *C. turicensis* [108]. This outer membrane protein (OMP) and others packaged within OMVs by *Cronobacter* were also confirmed by Kothary et al. [108] using PCR and DNA microarray analyses. *icsA*, which is harbored on plasmid pWR100, the virulence plasmid of *Shigella* species, is well known to encode for an autotransporter protein and is responsible for the intracellular/intercellular bacterial movement of this pathogen through polar deposition of filamentous actin to bind to the bacterial cell surface [109].

Bioinformatically, Grim et al. also found several autotransporter secretion gene loci (single genes or pairs of genes within the core genome) of *C. sakazakii* strain BAA-894, *C. malonaticus* strain LMG23826T, *C. turicensis* LMG 23827T, *C. dublinensis* subsp. *lactaridi* LMG 23825T, *C. dublinensis* subsp. *dublinensis* LMG 23823T, *C. dublinensis* subsp. *lausannensis* LMG 23824T, *C. muytjensii* ATCC 51329T and *C. universalis* NCTC 9529T, within accessory genomic regions 21 and 121 [102].

### 7.4. Type 6 Secretion System

The Type 6 secretion system is a secretion system which translocates effector proteins into host cells or into the environmental milieu using an outwardly directed phage-like structure for the secretion. Many of the effector proteins such as Hcp1 (hemolysin co-regulated protein 1) and VgrG (valine-glycine repeat G protein) are toxins that play a role in bacterial pathogenesis and environmental survival. This is accomplished through augmenting competition by subverting host-pathogen interactions away from pathogenesis and towards a more commensal or mutualistic state or it may mediate cooperative interactions between bacteria [110,111]. T6SS gene clusters have been found in many bacterial species [110]. Several T6SS gene clusters have been identified among the various *Cronobacter* species which are located both within the *Cronobacter* core genome as well as on a pESA3 virulence plasmid in *C. sakazakii* [47]. T6SSs are typically comprised of a conserved core gene cluster of up to 15 open reading frames (ORFs) [112]. Franco et al. reported in silico findings of a T6SS gene cluster consisting of 16 ORFs (ESA_pESA3p05491 to p05506) containing genes encoding for both Hcp1 and VgrG [47]. Other T6SS genetic components harbored on plasmid pESA3 included genes for IcmF-DotU/IcmH-SciN homologues which were found to share significant homology with T6SS stabilization proteins [112]. Additionally, within the plasmid pESA3 gene cluster is *clpV*, whose ATPase activity is crucial for T6SS activity [110]. The T6SS gene cluster contained on plasmid pESA3 is flanked upstream by a *gntR*-like homologue whose product is characterized as a transcriptional regulator [47]. Interestingly, downstream of the plasmid pESA3, T6SS gene cluster are composed of three putative genes encoding increased copper tolerance, such as *dsbG* [47]. The pESA3 regions flanking the T6SS gene cluster are conserved on plasmid pCTU1; however, the *C. sakazakii* T6SS gene cluster is replaced instead with a specific 32-bp sequence region. Recently, Wang et al. showed that *C. sakazakii* strain 12868 possessed two T6SS systems [113]. Their results suggest that T6SS-1 may contribute to interbacterial species competition processes which may allow *C. sakazakii* to better compete with other species in particular niches and the second gene cluster (T6SS-2) may be important during host interaction. Much more information is still needed regarding T6SSs.

## 8. Quorum Sensing Signaling Systems

Biofilm formation and expression of virulence factors in many bacteria have been found to be mediated by quorum sensing (QS) mechanisms [36]. Quorum sensing is a physiological cell-to-cell communication system involving the synthesis, secretion and subsequent detection of signaling molecules of low molar mass [36,114]. Increased extracellular concentrations of these signalling molecules are detected by cells and upon reaching a population density-dependent threshold induces specific, targeted, and coordinated expression of genes [114,115,116,117]. In Gram-negative bacteria, several structurally unrelated signal molecules have been identified including *N*-acyl-homoserine lactones (AHLs), alkylquinolones, a-hydroxyketones, diketopiperazines (DKPs) and small diffusable signal factors (DSFs) mimicking fatty acid compounds [30,118]. For a comprehensive review on the synthesis and characteristics of these molecules please refer to the review by Hawver et al. [118].

In a study by Lehner et al., the ability of *Cronobacter* species (reported as *E. sakazakii*) to form AHLs was investigated on a set of biofilm forming isolates using ethyl actetate extracts of cell-free supernatants [35]. The results indicated the presence of two different types of AHLs (3-oxo-C6-HSL and 3-oxo-C8-HSL) in these organisms. Pinton et al. also detected short chain acyl-HSL in *E. sakazakii* isolated from raw milk using a bioassay with *Chromobacterium violaceum* [119]. However, in this early study, the respective signaling molecules were not chemically characterized in detail.

Da Silva Araujo et al. examined short chain acyl-HSLs produced by *Cronobacter* species (reported as *E. sakazakii*) isolated from a feeding bottle, using mass spectroscopy [120]. The three molecules that were identified included (*S*)-*N*-heptanoyl-HSL, (*S*)-*N*-dodecanoyl-HSL, and (*S*)-*N*-tetradecanoyl-HSL. In that same study, it was reported that *Bacillus cereus* was capable of secreting two exoproteins, an acyl-HSL lactonase and AHL acylases which resulted in depletion of the acyl-HSL secreted by *Cronobacter* species [120]. This interference in quorum sensing mechanisms, also known as quorum quenching, has been proposed as a promising alternative to control bacterial virulence. However, as observed in this later study, the depletion of the *E. sakazakii* acyl-HSLs by *B. cereus* extracelluar enzymes did not inhibit the growth or biofilm formation of *E. sakazakii*, suggesting the presence of alternative signaling molecules for both the *E. sakazakii* and the *B. cereus* QS and QS quenching systems or suggests that signaling is not critical in those assays.

Indeed, a very recent study showed that an alternative type of QS system exists, a cyclo (L-Pro-L-Leu) diketopiperazine was detected in pure and mixed cultures of *C. sakazakii* and *B. cereus*, possibly acting as cross-communication QS signals between these two organisms [121]. 2,5-DKPs are QS molecules commonly found in Gram-positive bacteria and are not usually secreted by Gram-negative microorganisms. However, previous studies showed that DKPs can modulate gene expression controlled by the expression of key regulatory proteins, substituting AHLs in a Gram-negative bacterium’s QS scheme [122,123]. DKPs and short chain AHLs bind to the same regulatory protein allowing the co-sharing of an econiche by different microbial genera/species [123].

In 2016, Suppiger et al. reported that *Cronobacter* species secreted a diffusable signal factor (DFS) which was synthesized by DFS-type quorum sensing system [39]. Expression of this system was involved in the regulation of several phenotypes, including biofilm formation, colony morphology and swarming motility. Knock-out mutants of the sensing (RpfF) and the responding (RpfR) signal coding genes in *C. turicensis* strains were used in a Zebrafish embryo model and it showed a role of this regulatory system in the virulence of *C. turicensis* [39]. In addition, the study provided evidence that the RpfF/R system modulates the intracellular cyclic-di-GMP levels within the organism, indicating that this secondary messenger is important in virulence and in regulating the expression of the above phenotypes.

In another recent study, long chain AHLs (C6–C18 in length) were identified and chemically characterized in *C. sakazakii* [124,125]. In vitro results demonstrated that these AHLs appeared sufficient to be detected after 6 h of incubation [126]. These authors showed that strains, which secreted significant levels of these AHLs, also produced significantly more extra-cellular polysaccharides (EPS) and formed more biofilms [126]. Unfortunately, data on the global regulatory circuitry that may be involved or other pleotropic effects possibly triggered through this QS system were not provided in this study [126].

In order to obtain a more comprehensive picture of the genes, secreted exoproteins, and regulatory mechanisms of these cell-to-cell communication systems, a more global approach such as transcriptomic or proteomic analyses are warranted. Such studies will identify possible secreted exoproteins in *Cronobacter* species that may also act as toxins. In addition, secreted exoprotein molecules that interfere with QS signaling may also provide an alternative approach to control *Cronobacter* organisms in dairy food production environments, as well as in disease. Though knowledge of how QS fundamentally controls *Cronobacter* virulence is in its early stages, some inferences can be made based on known common themes of QS regulation, such as the LuxR/SmcR regulatory schemes found among *Vibrio* species [125].

## 9. Outer Membrane Proteins (OMPs)

Although the genetic, genomics, and transcriptomics of *Cronobacter* pathogenesis has not been fully understood, a greater amount of information has been collected about several virulence factors, such as toxins, and putative genetic loci that may contribute to pathogenesis. Two well characterized outer membrane-related virulence genes, *ompA* and *ompX* (encoding for outer membrane proteins A and X) are involved in adherence, invasion of intestinal cell epithelial, and brain endothelial cells through bacterial cell binding of host fibronectin [75]. Kothary et al. showed that both OmpA and OmpX are packaged within OMVs, which are expressed in stationary phased-grown cells of *C. sakazakii*, *C. malonaticus* and *C. turicensis* [108]. Ye et al. showed that the expression of these OMPs were greater in a virulent *C. sakazakii* strain than that of an avirulent strain [127]. Kothary et al. also found that other OMPs were packaged within secreted OMVs and include MipA (peptidoglycan synthesis), porin proteins (OmpC, OmpE, and OmpF), a conjugative plasmid-T4SS protein (CTP), a chaperonin (GroEL), and the previously mentioned OM autotransporter protein (OMATP) [93]. It is thought that these OMPs were packaged in a purposeful way and not randomly, and since the cells used to harvest the OMVs were in stationary phase of growth, these OMPs represent proteins expressed by cells grown under stress. Furthermore, parallel PCR and microarray analyses of 240 strains representing the seven *Cronobacter* species showed that these OMPs are highly conserved among all the species [108]. OMVs were once thought to be cellular artifacts, but currently, OMVs are now acknowledged structures possessing very diverse functions and are currently regarded as a protein secretion system used by bacteria to communicate with host cells and other bacterial cells. In addition to OMPs, Kothary et al. also showed that these OMVs contained substantial amounts of LPS [108]. The existence of LPS components in PIF has been known [128] and may lead to a situation where the LPS in PIF causes an unrestrained pro-inflammatory response in a susceptible host which could lead to a fatal “cytokine storm”. It was shown by Townsend et al. [128] that the presence of LPS along with *C. sakazakii* in infant formula augmented the translocation of *C. sakazakii* from the rat gut lumen to the meninges through translocation of the blood-brain-barrier. One explanation is that the permeability of the host barrier to the pathogen was increased through disruption of cellular tight junctions by LPS [32]. More information is required to extend our understanding of the process regulating the production of OMVs and associated OMPs and their roles in disease.

## 10. Exoproteins Involved in Osmotic Stress Response

It is thought that the *Cronobacter* genus split from its most recent *Enterobacteriaceae* ancestor approximately 45–68 million years ago. This time period coincides with the Paleogene period of the Cenozoic era when early flowering plants were also evolving, thus supporting the theory that plants may be the ancestral habitat for *Cronobacter* species [56,103,129,130]. Recently, Afridi et al. provided evidence that plants inoculated with *C. sakazakii* (producing 1-aminocyclopropane-1-carboxylate deaminase, ACC deaminase) enhanced plant growth and saline stress tolerance [131]. It was concluded that ACC utilization by *C. sakazakii* promoted plant growth due to the lowering of excess ethylene production under salt stress. Additionally, it has been reported that *Cronobacter* have been isolated from flies [24]; thus, it is conceivable that the feeding of insect larvae on plants could have led to colonization of the fly and subsequent host adaptation and the further evolution of the genus [30]. Interestingly, *Kocuria rhizophila*, another ACC producing bacterium was isolated from the gut of pine lappet moth (*Kunugia latipennis*) [131]. During the Cenozoic era, it is also thought that *Cronobacter* species acquired the capacity to endure low-moisture environments thereby contributing to increased survival and persistence in low water activity food matrices and in their associated production environments. Little is known about how *Cronobacter* survive and persist in these low-moisture environments. Therefore, a complete understanding of stress adaptation is imperative to facilitate the design of strategies to mitigate its survival in PIF and other low-moisture food matrices. Bacterial osmotic stress responses (e.g., growth of cells in varied physiological environments of extreme salinity and/or osmolarity) are complex and involves both primary and secondary responses [132,133,134]. These cellular responses must be interpreted as a distinct sequence of cellular events, which have been well characterized in *E. coli* [132]. Generally, exposure of cells to high external osmolar growth conditions causes an efflux of water from the cell interior, resulting in a reduction of turgor pressure. Furthermore, an increase in the concentrations of intracellular metabolites and ions occurs. This is counterweighed by active potassium efflux and glutamate synthesis (the mainstay of the primary response) to restore intracellular water levels and, finally the potassium/glutamate is replaced with the accumulation of osmoprotectants, which is more compatible with cell growth (“compatible solutes”, a hallmark of the secondary response). In addition, high osmolarity growth conditions cause a rapid increase in negative DNA supercoiling, which may control transcription of osmoregulated genes such as the induction of enzymes needed for the elimination of oxygen radical species in response to heat or osmotic shock [133]. Riedel and Lehner revealed that expression of most of the proteins, that were upregulated in cells (*C. sakazakii* strain 236) grown under desiccation stress, were either outer membrane proteins (e.g., OmpC and A) or proteins involved in transport of inorganic ions and energy production such as ATPases [133]. Other proteins found included Clp, chaperonin GroES (chaperones), Gln-binding periplasmic protein that are involved in amino acid transport and metabolism, an enolase, the PTS system, a glucose-specific IIA component, α-Glucosidase (carbohydrate transport and metabolism), and inorganic pyrophosphatase (energy production and conversion) [133]. Interestingly, Feeney and Sleator also described a comparative genomic approach to investigate the ability of *C. sakazakii* to survive and persist under low water activity growth conditions [132] and these authors identified fifty-three genes that were involved in osmotolerance, including those associated with both hyper- and hypo-osmotic stress response systems. Various homologues of the principal osmotolerance genes of *E. coli* were also found; however, a key difference noted between *C. sakazakii* and *E. coli* was that *C. sakazakii* contained multiple copies of *proP* (seven) and two copies of *opuC*, which is involved in carnitine uptake, and has also been found to transport other osmoprotectants or solutes such as glycine betaine, proline, ectoine and choline [132]. Furthermore, it was noted that the osmotic stress response of *C. sakazakii* appeared to be regulated at the transcriptional, translational and post-translational levels, and these researchers suggested that RpoS most likely be functioning as a global transcriptional regulator involved in the osmotolerance response. A recent report describing RNA-sequence data obtained from cells of desiccated *C. sakazakii* strain SP291 shows that about 25% of the total *C. sakazakii* genes were significantly up-regulated and 10% of the genome were down-regulated [134]. qRT-PCR analysis demonstrated that the primary desiccation response involved the rapid accumulation of potassium glutamate to provide temporary protection against desiccation stress by immediately increasing the internal osmotic pressure of the bacterial cell [134]. This response was gradually downregulated over time while the secondary response was found in desiccated SP291 cells to remain constitutively up-regulated throughout the experiment. The trehalose biosynthetic pathway encoded by *otsA* and *otsB*, are prominent secondary participants and were highly up-regulated in these desiccated *C. sakazakii* cells. Knockout mutants (deleted in *otsAB)* yielded considerable inhibition of desiccation survival compared to the isogenic wild type, confirming the physiological significance of trehalose in desiccation survival, but survival was not absolutely abolished, signifying that other unknown factors may be involved in the response to desiccation [134].

Besides the ability to tolerate, persist, and survive under high osmotic stress, *Cronobacter* have also adapted quite well to exposure to high growth temperature conditions [23,135]. Additionally, Williams et al. identified a hypothetical protein in several highly heat-tolerant *Cronobacter* strains which shared sequence homology with a protein found in the thermal tolerant bacterium, *Methylobacillus flagellatus* KT [136]. Furthermore, Gajdosova et al. [137] described an 18-kbp gene region that contained a cluster of genes, including a thermal tolerance protein described by Williams et al. [136] that had significant homology with other known bacterial proteins involved in many types of stress responses including heat, oxidation and acid stress. This island (called TPQLC or LHR) is found on the chromosome but appears to be horizontally acquired as it is flanked by transposases. However, not every strain that possesses a thermotolerant phenotype was found to possess this gene cluster [138], suggesting that other molecular mechanisms of thermotolerance exists. Yan et al. also found that *C. sakazakii* strain SP291 did not possess the *M. flagellatus* KT thermotolerance marker [138]. However, SP291 does have a shortened version of the TPQLC/LHR island (~6.5 kb) [138]. More importantly, the truncated SP291 TPQLC/LHR island encodes the small heat shock protein and ClpG, which has been identified as the main source of heat resistance. Furthermore, Mercer et al. found evidence that the LHR island and its heat resistance function is conserved in many *Enterobacteriaceae* members [139].

## 11. Efflux Pumps

Efflux or transport of molecules across the Gram-negative bacterial cell envelope can be achieved in a single energy-coupled step. This “transport apparatus”, is called an efflux pump (EP), which allows for nonstop passage of molecules across both inner and outer membranes and the intervening periplasmic space (does not generate periplasmic intermediates). In contrast to EPs, other protein secretion processes involve step-wise translocation through the inner cell membrane and the outer cell envelope via the periplasmic space [140]. Efflux pumps also can interact with different translocase complexes, depending on the transported substrate, so that these protein complexes provide the cell a greater functional diversity in secretion. In fact, many of these efflux pumps are multidrug transport efflux systems and many interconnect with other OMPs such as TolC, which belongs to a family of OMPs found in all Gram-negative bacteria. These proteins are essential for the expulsion of a plethora of compounds such as potentially small lethal agents, such as detergents, solvents, heavy metals and antibiotics [140]. In addition, there is accumulating evidence that efflux pumps that confer clinically relevant antibiotic resistance are also important in virulence; for example, the phytopathogen, *Ralstonia solanacearum* causes wilt disease in tomatoes [141]. In this study, mutants deficient in *acrA* and *dinF* genes, were significantly less virulent to tomato plants than the wild type strain. Another example of efflux pumps contributing to bacterial virulence is with *Vibrio cholerae*, the etiological agent of cholera. *V. cholerae* El Tor strain N16961 possesses six genes encoding for resistance-nodulation-division (RND) efflux pumps [142]. Mutants deficient in these efflux pump genes, including an RND-null strain, produced significantly less cholera toxin and fewer toxin-co-regulated pili than the wild type strain and was unable to colonize the infant mouse [142]. It was also found that a decreased virulence factor production in the RND-null strain was also linked to reduced transcription of *tcpP* and *toxT,* two membrane-associated transcriptional activators which directly activate transcription of the cholera toxin and toxin-coregulated pilus genes [142].

Negrete et al. found 13 different families of efflux pumps (shown in Table 2) among the seven *Cronobacter* species [143]. These efflux pumps were found associated with specific genomic regions (GR) such as gene clusters involved in sugar transportation, heavy metal efflux systems (Arsenic, copper efflux on plasmids pSP291-2, pCS2, and pCTU3). Interestingly, searching the chromosome of *C. sakazakii* strain BAA-894, 24 efflux-related genes were found that encode for RND efflux pump proteins involved in the transport of multidrug, and copper efflux (CusA/CzcA family heavy metal efflux), a MATE family efflux transporter, MFS Transport (EmrB and Bcr/CflA), glutathione-regulated potassium-efflux (KefC, KefF, KefG, and KefB) to name a few efflux systems [143]. Two efflux genes encoding for an efflux RND transporter periplasmic adaptor subunit and an arsenical efflux pump membrane protein ArsB, were found located on the virulence plasmid, pESA3 [143].

## 12. Role of Plasmids

Several plasmids of various sizes ranging from 4.4 kbp to 197.3 kbp have been described for *Cronobacter* species and a summary of these plasmids is shown in Table 2 [33,47,52,53,144,145]. pESA3-like plasmids (synonymous with plasmids pCTU1, pCS2, pCSK29544_1, and pSP291-1) are thought to be examples of a prototypic virulence plasmid that was originally described by Franco et al. [47]. Plasmid pEAS2 is similar to pCTU2, which are conjugative plasmids, and pCTU3 is a plasmid that contains gene clusters involved in heavy metal (Ag, Cu, and Arsenic) efflux (efflux pumps such as RND EPs); and maintenance of these plasmids seem to be under tight control by multiple toxin-antitoxin genes of both type I and type II classes of toxin-antitoxin (TA) genes [46,52,53,102,145]. TA genes are thought to help stabilize plasmids and mobile genetic elements or genetic cassettes, and they participate in the response to stressful growth conditions. Activation of TAs in response to stress is thought to control the metabolic load of cells within a population by eliminating part of the population (through outer membrane lysis); this is followed by the surviving cells entering a physiological resting or dormant state [146]. Interestingly, a type II toxin-antitoxin system HipA family toxin gene (WP_041460783) is located just downstream of the *iucABCD/iutA* Cronobactin siderophore and the TonB-dependent siderophore receptor genes (WP_041460784.1) [46,47]. There are also chromosomally located TAs which have been shown to control several bacterial processes, like biofilm formation, survival during infection of eukaryotic cells, defense against invading bacteriophages and entrance and exit into persistence [146].

The role of pESA3/pCTU1 plasmids in virulence was confirmed by Eshwar et al. [76], who showed that strains harboring plasmids pESA3 and pCTU1 exhibited twice the mortality rate than isogenically plasmid-cured strains or naturally occurring plasmid-free strains using the Zebrafish infection model. These data suggest that these plasmids are virulence-associated but may not represent the entire virulence factor gene repertoire of *Cronobacter*. pESA3/pCTU1-like plasmids encode a single RepFIB-like origin of replication gene, *repA*, as well as genes for two iron acquisition systems (*eitCBAD* and *iucABCD/iutA*). pESA3-like plasmids possessed by *C. sakazakii* also harbor a type VI secretion system that controls the secretion of toxic proteins such as Hcp1 and VrgR [47]. Table 1, Table 3 and Table 4 describe the various loci carried on these plasmids in greater detail as harbored by the various *Cronobacter* species. Table 1 and Table 4 show the prevalence and distribution of the pESA3/pCTU1-like virulence plasmids compared to pESA2/pCTU2-like conjugal and pCTU3-like plasmids among 570 *Cronobacter* strains. Table 3 describes the genomic characteristics of completely sequenced plasmids carried by the seven *Cronobacter* species.

## 13. Role of Secreted and Assembled Cell Surface Proteins (Adherence Factors) in Disease and Persistence

It is generally considered that the ability of pathogenic bacteria to adhere to an epithelial cell surface is the first step in pathogenesis [148]. Bacteria express filamentous assemblies of protein subunits called pili or fimbriae which are used to colonize a host cell membrane surface or as conduits for the secretion of substrates (e.g., T4SS fimbriae). These adherence factors are proteinaceous assemblies that extend from the cell surface and are secreted and assembled by either a chaperone/usher-dependent or a structural subunit/nucleator-precipitation pathway [149]. Fimbriae facilitate adherence of bacterial cells to host tissue cells through the interaction with host receptors located on surface of the target cell. These interactions are often tissue specific, which occur either with the fimbriae main structural subunit or with associated fimbrial adhesins and arise through recognition by the fimbrial adhesin of certain chemical groups of host ligands (various glycosylated membrane receptors) [149]. The genetic loci coding for these structures are found both on the chromosome and on plasmids [149].

Grim et al. [102] described eight fimbriae types in the seven *Cronobacter* species that were based on the chaperone-usher classification system described by Humphries et al. [150]. These included fimbriae identified as γ1, γ4, κ, β, π, and Ʃ types, curli, and a P-pilus homologue, that were differentially dispersed among the *Cronobacter* genomes analyzed in their study [102]. Some genomes also harbored curli biosynthesis genes, which are homologous to curli of *E. coli* and thin-aggregative fimbriae of *Salmonella* [149,150]. Curli fimbriae belong to a type of highly aggregated surface protein fibers (6–8 nm in diameter and 1 μm in length) that are related to proteins called amyloids and are involved in adhesion to other cells or material surfaces. They have also been found to be involved in cell-cell aggregation, and biofilm development [150]. The biosynthesis of curli is encoded by two operons, *csgBAC* and *csgDEFG* (csg, curli-specific genes in *E. coli*) [150]. The major curlin structural subunit encoded by *csgA* and *csgB* encodes a nucleator protein subunit while *csgC* may have accessory function in the formation of curli fimbriae [141]. Curli fimbriae are not assembled if CsgB is absent, so CsgA is secreted from the cell as an unpolymerized exoprotein [150]. Using primers designed to detect the structural curlin subunit gene (*csgA*) and a putative assembly factor gene (*csgG*), Hu [151] found that *csgA* was found in *C. dublinensis*, *C. malonaticus*, *C. turicensis* and *C. universalis*, but not in *C. sakazakii* and *C. muytjensii*. Using the PATRIC tool and NCBI’s genome protein tables, Jang et al. [88] showed that the prevalence and distribution of Type 1, Beta, Sigma, Pap, and Curli fimbriae gene clusters possessed by the seven *Cronobacter* species followed species lines and are summarized in Table 5. For example, Beta fimbriae were only seen in *C. sakazakii* strains and Curli fimbriae were not found in *C. sakazakii* nor *C. muytjensii* strains. Interestingly, *C. muytjensii* strains also did not possess genes for Sigma fimbriae (Table 5).

## 14. Iron Transport

Iron is used as a cofactor in many essential enzymes involved in basic cellular functions, which are also associated with metabolic pathways of both pathogens and their hosts [153,154]. Iron is found in two forms, Ferric (Fe^+3^) and Ferrous (Fe^+2^) iron. Within the host, Fe^+3^ iron is usually only available when it is bound to specific proteins, such as transferrin, lactoferrin, and ferritin, or when it is complexed with hemoproteins [153,154]. The Ferric (Fe^+3^) storage form must be reduced to (Fe^+2^) to cross a plasma membrane. During evolution and associations with their various ancestral hosts, bacteria have developed several mechanisms to utilize indigenous stores of host iron. Specialized iron-uptake systems have been found in most bacterial species, and these systems allow microorganisms to compete for iron within hosts or as members of mixed microbial environmental communities. As mentioned earlier, Franco et al. found that pESA3/pCTU1-like plasmids encode common virulence factors, including a hydroxamate-type or aerobactin-like siderophore named cronobactin (*iucABCD* and the receptor gene *iutA*) and an ABC ferric-iron transporter gene cluster *eitABCD* [47]. In Gram-negative bacteria, siderophores are specialized iron binding ligands, which sequester the iron and subsequently interact with siderophore receptors and an ATP-driven porin-like (TonB- like) transporter protein located in the bacterial outer membrane.

In addition to the plasmid-borne iron acquisition gene clusters, *Cronobacter* spp. have other transport systems to transport both ferric and ferrous iron. These systems include genes encoding ferric and ferrous transporters and heme-iron extractors, as well as putative TonB-dependent iron receptors and ferric reductases [102]. For acquisition of ferrous iron, all *Cronobacter* species have two ferrous iron transporters systems (Feo and Efe), and for transport of ferric iron, all plasmid-harboring strains (97%) have the siderophore cronobactin [102]. *Cronobacter* species also harbor genes homologous to the *fhuACDB* operon of *E. coli* which suggests that, like *E. coli*, *Cronobacter* species can scavenge iron from siderophores produced and secreted by other microorganisms [102]. Interestingly, phylogenetic analysis of most of the iron acquisition genes and systems separate the genus *Cronobacter* into two subclades: one subclade includes the species *C. sakazakii*, *C. malonaticus*, *C. universalis*, and *C. turicensis*, and the other subclade is comprised of *C. muytjensii* and *C. dublinensis* which is similar to that described by Grim et al. [102] for members of the genus. For a comprehensive review on the synthesis and characteristics of the various iron transport systems and associated proteins that *Cronobacter* species possess, please refer to a paper by Grim et al. [102].

## 15. Conclusions

*Cronobacter* species and the diseases they cause have undergone a significant adjustment since 2008 when Iversen et al. [1] and Joseph et al. [2] reclassified the genus into its current seven species; and undoubtedly this taxonomic scheme will continue to evolve. Once thought to be only a harmless inhabitant of the intestinal tract of humans, *Cronobacter* species are now considered to be a group of pathogens with notable versatility in their ability to cause human disease in all age groups. However, neonates and infants are still regarded as the age groups that are highly susceptible to invasive disease and disease susceptibility can now be extended to adults, primarily elderly individuals. Various species-specific virulence factors have been described that can adversely affect a wide range of eukaryotic cell processes including protein synthesis, cell division, and ion secretion. Many of these factors are toxins that have been shown to also modulate the host immune system. These factors are also encoded on a variety of mobile genetic elements such as plasmids and transposons; this genomic plasticity implies ongoing re-assortment of virulence factor genes and will undoubtedly continue to complicate our efforts to categorize *Cronobacter* into sharply defined genomic pathotypes. Furthermore, much information regarding its association with the postpartum intestinal microbiota is also warranted so to determine if a human carriage state is involved in the transmission of *Cronobacter* from infant caretakers to susceptible infants. Lastly, more in-depth surveillance studies using species-specific identification methodologies that at least include results of species-specific PCR assays, or at best, species identities obtained from next generation sequencing (NGS) studies are needed to obtain a clearer epidemiologic picture as to which *Cronobacter* species are responsible for infections. Accordingly, the collective suggestion of public health and food safety officials is to elevate *Cronobacter* species to the level where it should be mandatorily submitted to the various centralized disease reporting systems as suggested by Tall et al. [147]. Two such systems are the National Notifiable Diseases Surveillance System, which is maintained by the Centers for Disease Control and Prevention (Available online: https://wwwn.cdc.gov/nndss/data-and-statistics.html, last accessed 1/17/2020) and the European Centre for Disease Prevention and Control’s Surveillance System (Available online: TESSy, https://ecdc.europa.eu/en/publications-data/european-surveillance-system-tessy, last accessed 1/17/2020) [20,155]. Currently, Minnesota is the only state in the USA that does this. In summary, many of these bacterial factors described in this review are toxins or exoproteins which have been shown to also modulate the host immune response. Many of the genes for these proteins are encoded on a variety of mobile genetic elements such as plasmids; this genomic plasticity seems to be common among the seven species and also indicates ongoing re-assortment of future virulence factor genes. Such genomic reassortments have complicated efforts to categorize *Cronobacter* into sharply defined genomic pathotypes. As the use of WGS increases, it is hoped that the finding of new genomic attributes will allow for a better understanding of virulence. Combining this information with an improved epidemiological reporting system will lead to a more comprehensive understanding of toxin secretion and virulence. The ultimate goal would be improved patient care enabling better clinical outcomes.

## Figures and Tables

**Figure 1 microorganisms-08-00229-f001:**
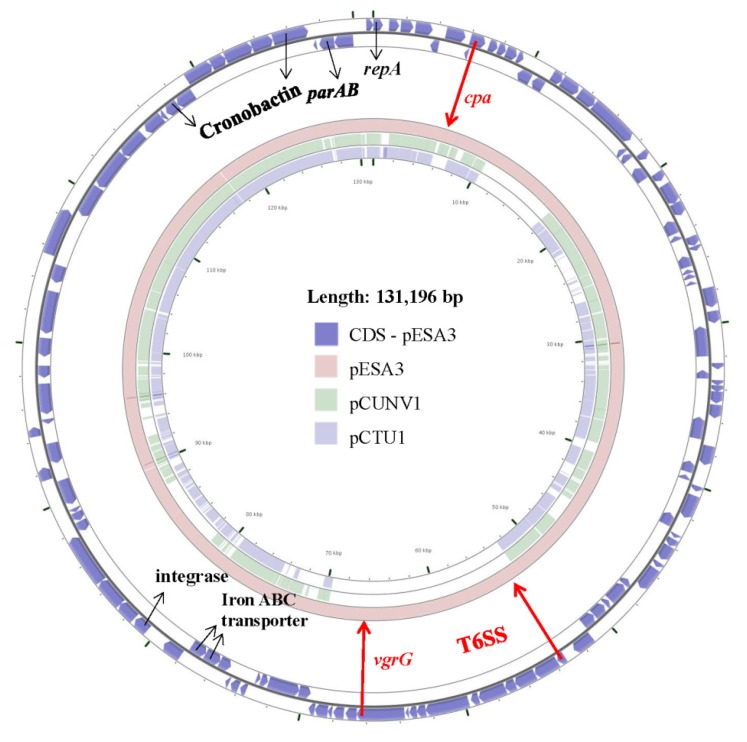
Sequence alignment of pESA3, pCS2, pSP291–1 and pCTU1 produced on the CGView Server from the Stothard Research Group [55] that uses BLAST analysis to illustrate conserved and missing genomic sequences (Available online: http://stothard.afns.ualberta.ca/cgview_server/; last accessed 12/20/2019). Two circular plasmid genomes, pCUNV1 (NZ_CP012258) and pCTU1 (NC_013283), were compared against the reference pESA3 (NC_009780). GenBank annotations of the reference pESA3 (CDS in blue arranged in two outside rings) were downloaded as a GFF file for analysis using the default configuration on the CGView server. Select genes or loci of interest are shown as across the circular genomes as follows: Siderophore loci with Cronobactin gene, Iron ABC transporter genes, Type 6 Secretion System (T6SS), *parAB* genes and the toxin *cpa* gene are adapted from Franco et al. [47]. Missing regions identified by the BLAST analysis on the CGView server are shown as ‘gaps’ on each of the two circular genomes. Genes and loci missing in pCUNV1 or pCTU1 plasmids are in red. As expected, T6SS is seen only on the reference pESA3 from *C. sakazakii* while the toxin encoding *cpa* gene is absent in the plasmid pCTU1 from *C. turicensis.* Figure was adapted from Jang et al. [56].

**Figure 2 microorganisms-08-00229-f002:**
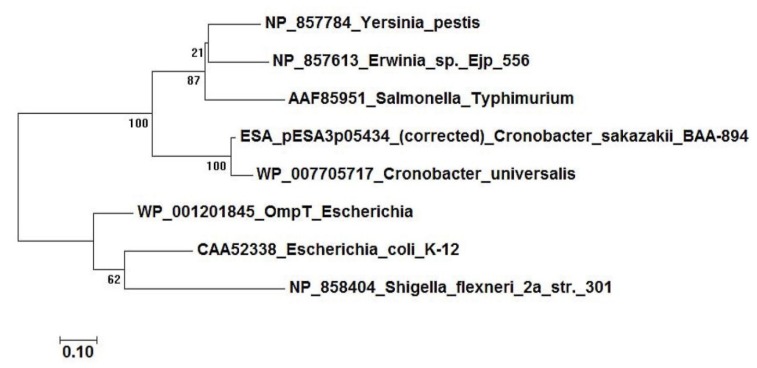
Phylogenetic tree of the homologs of omptin, Cpa. The NCBI accession numbers of the proteins sequences used in the figure are as follows: *Yersinia pestis,* Pla (plasminogen activator, NP_857784); *S. enterica* Typhimirium, PgtE (outer membrane serine protease, AAF85951); *Erwinia*, PlaA (plasmid, NP_857613); *C. sakazakii* BAA-894, Cpa (plasmid, ESA_pESA3p05434); *C. universalis* NCTC 9529, Cpa (omptin family outer membrane protease, WP_007705717); *E. coli*, OmpT (outer membrane protease VII, AP_001210); *E. coli,* OmpP (outer membrane protease P, X74278); and *Shigella flexneri*, SopA (outer membrane protease, NP_858404). Forty-one amino acids were added to *C. sakazakii* Cpa protein in its N-terminal to correct the incomplete annotation of the protein in the GenBank record. The MUSCLE algorithm of the MEGA7 suite was used for multiple sequence alignment. Phylogenetic analyses were conducted in MEGA7 using the Maximum-Likelihood algorithm [77]. Three hundred nine amino acid positions across the protein were used for determining the distance between the homologs in the tree. Bar marker represents 0.1 amino acid differences. Confidence values given in the nodes were derived out of bootstrap test consisting of 500 iterations.

**Figure 3 microorganisms-08-00229-f003:**
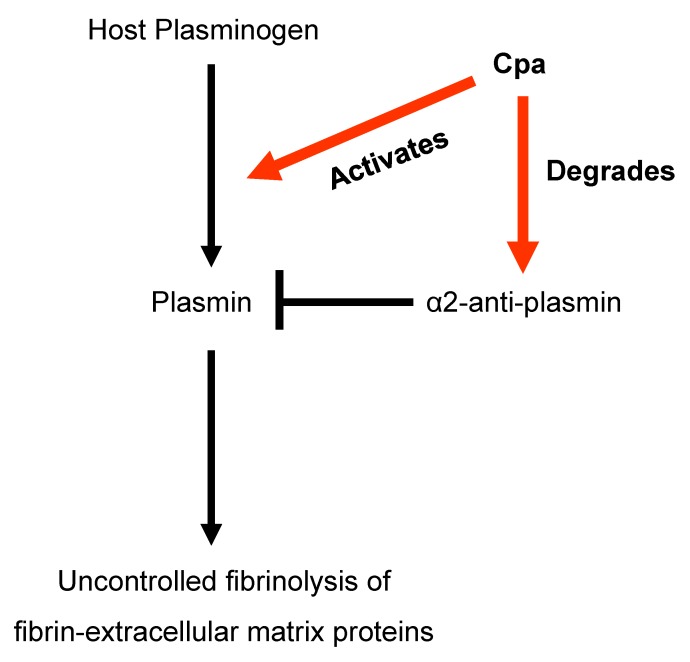
Mechanisms of plasminogen activation by *C. sakazakii* and its role in bacterial virulence. It is thought that a complex with plasminogen is formed when *Cronobacter* plasminogen activator (Cpa) is expressed by invasive *C. sakazakii* (cells invading a host’s circulatory system), which causes proteolysis and conversion of host plasminogen to plasmin. Plasmin bound (conjecture) to bacterial cell surfaces then catalyzes the degradation of fibrin polymers (fibrinolysis) which are major components of fibrin clots and the extracellular matrix. Additionally, Cpa can also inactivate α2-anti-plasmin which normally would break down plasmin. Thus, there is an unlimited activation of plasmin leading to increased fibrinolysis which in turns allows for increased invasiveness.

**Table 1 microorganisms-08-00229-t001:** Comparison of prevalence and distribution of pESA3/pCTU1 (incFIB), pESA2/pCTU2 (incF2), and pCTU3 (incH1) plasmids among 570 *Cronobacter* isolates.

Species	No. of Isolates	No. of Isolates with the Indicated Plasmid Incompatibility Class (%) ^a^		
		pESA3/pCTU1 (incFIB) ^b^	pESA2/pCTU2 (incF2)	pCTU3 (incH1)
*C. sakazakii*	507	493 (97)	20 (4)	142 (28)
*C. malonaticus*	30	30 (100)	3 (10)	12 (40)
*C. turicensis*	13	13 (100)	1 (8)	8 (62)
*C. muytjensii*	12	9 (75)	0 (0)	1 (8)
*C. dublinensis*	5	4 (80)	0 (0)	0 (0)
*C. universalis*	2	2 (100)	0 (0)	1 (50)
*C. condimenti*	1	1 (100)	0 (0)	0 (0)
Total	570	552 (97)	24 (4)	164 (28)

^a^ Numbers within parentheses are the percentage PCR-positive for each plasmid replicon gene locus (*repA*) as described by Franco et al. [47]. The prevalence of pESA3/pCTU1 (incFIB), pESA2/pCTU2 (incF2), and pCTU3 (incH1) plasmids among the strains were calculated using the total number of strains tested. ^b^ The column labelled pESA3/pCTU1 (incFIB) is also presented in Table 4 as well for ease of comparison of the prevalence and distribution among the three plasmid types of the seven *Cronobacter*.

**Table 2 microorganisms-08-00229-t002:** Prevalence and distribution and NCBI protein annotations of efflux pump associated genes among the seven *Cronobacter* species as described on the pan-genomic *Cronobacter* microarray ^a^.

Family	Sub-Cluster	NCBI Efflux Protein	*C. sak*	*C. tur*	*C. dub*	*C. con*	*C. mal*	*C. muy*	*C. uni*
Kef	Potassium efflux system KefA protein / Small-conductance mechanosensitive channel	ABU75471	+	+	+	+	+	+	+
	Potassium efflux system KefA protein / Small-conductance mechanosensitive channel	ABU77777	+	+	+	+	+	+	+
	Potassium efflux system KefA protein / Small-conductance mechanosensitive channel	ABU78035	+	+	+	+	+	+	+
	Glutathione-regulated potassium-efflux system protein KefB	ABU79568	+	+	+	+	+	+	+
	Glutathione-regulated potassium-efflux system protein KefC	ABU78514	+	+	+	+	+	+	+
	Glutathione-regulated potassium-efflux system ancillary protein KefF	ABU78515	+	+	+	+	+	+	+
	Glutathione-regulated potassium-efflux system ancillary protein KefG	ABU75563	+	+	+	+	+	+	+
	Glutathione-regulated potassium-efflux system ancillary protein KefG	ABU79567	+	+	+	+	+	+	+
	Glutathione-regulated potassium-efflux system ATP-binding protein	ABU76397	+	+	+	+	+	+	+
	Putative metal-binding cytoplasmic protein probably associated with glutathione-regulated potassium-efflux	ABU79569	+	+	+	‒	+	+	+
RND	Membrane fusion protein of RND family multidrug efflux pump	ABU78037	+	+	+	+	+	+	+
	Membrane fusion protein of RND family multidrug efflux pump	ABU78865	+	+	+	+	+	+	+
	Probable RND efflux membrane fusion protein	ABU76411	+	+	+	+	+	+	+
	Cation efflux system protein CusC precursor	ABU79419	+	+	‒	‒	+	‒	‒
	Cobalt-zinc-cadmium resistance protein / heavy metal efflux pump, CzcA family	ABU79422	+	+	‒	‒	+	‒	‒
	Cation efflux system protein CusF precursor	ABU79420	+	+	‒	‒	+	‒	‒
Threonine	Putative threonine efflux protein	ABU75741	+	+	+	‒	+	+	+
	Threonine efflux protein	ABU78513	+	+	+	+	+	+	+
PET	Putative efflux (PET) family inner membrane protein YccS	ABU77634	+	+	+	+	+	+	+
TetR (AcrR)	Transcription repressor of multidrug efflux pump acrAB TetR (AcrR) family	ABU78036	+	+	+	+	+	+	+
	Transcription repressor of multidrug efflux pump acrAB TetR (AcrR) family	ABU78864	+	+	+	+	+	+	+
Plasmic E.P	Periplasmic component of efflux system	ABU76384	+	‒	‒	‒	‒	‒	‒
	Outer membrane efflux family protein	ABU76385	+	‒	‒	‒	‒	‒	‒
CorC	Magnesium and cobalt efflux protein CorC	ABU76695	+	+	+	+	+	+	+
	Magnesium and cobalt efflux protein CorC	ABU77911	+	+	+	+	+	+	+
Mac	Macrolide-specific efflux protein MacA	ABU77706	+	+	+	+	+	+	+
DMT	Putative DMT superfamily metabolite efflux protein precursor	ABU77772	+	+	+	+	+	+	+
MFP	Predicted membrane fusion protein (MFP) component of efflux	ABU77795	+	+	+	+	+	‒	+
Lactone	Homoserine/homoserine lactone efflux protein	ABU78933	+	+	+	+	+	+	+
MATE	Multi antimicrobial extrusion protein (Na(+) driven), MATE family of MDR efflux pumps	ABU77280	+	+	+	+	+	+	+
Transporter	RND efflux system aminoglycoside inner membrane transporter CmeB	ABU76058	+	+	+	+	+	+	+
	Sugar efflux transporter B	ABU76341	+	+	+	+	+	+	+
	Formate/nitrite efflux transporter (TC 2.A.44 family)	ABU77686	+	+	+	+	+	+	+
	ABC transporter multidrug efflux pump fused ATP-binding domains	ABU77796	+	+	+	+	+	‒	+
	RND efflux system, inner membrane transporter CmeB	ABU78038	+	+	+	+	+	+	+
	Cobalt/zinc/cadmium efflux RND transporter membrane fusion protein, CzcB family	ABU79421	+	+	‒	‒	+	‒	‒
Unknown	Possible efflux pump	ABU78153	‒	+	+	+	+	+	+

^a^ Gene and NCBI annotation are adopted and described by Negrete et al. [143].

**Table 3 microorganisms-08-00229-t003:** Characteristics of known plasmids possessed by *Cronobacter* species ^a,b^.

*Cronobacter* Species and Strain	Plasmid Name	RefSeq	INSDC	Size (Kb)	GC (%)	Protein	Gene	Pseudogene	Reference
*C. universalis* NCTC 9529	pCUNV1	NZ_CP012258.1	CP012258	129.8	57.0	118	119	1	46
*C. sakazakii* 29544	CSK29544_1p	NZ_CP011048.1	CP011048	93.9	57.0	61	69	8	unpublished
*C. sakazakii* 29544	CSK29544_2p	NZ_CP011049.1	CP011049	4.9	54.9	4	7	3	unpublished
*C. sakazakii* 29544	CSK29544_3p	NZ_CP011050.1	CP011050	53.5	50.1	58	61	3	unpublished
*C. sakazakii* 29544	pCSA2	NC_021293.1	KC663407	5.1	55.0	6	6	0	unpublished
*C. sakazakii* NCIMB 8272, NCTC 8155	pCS1	NZ_CP012254.1	CP012254	110.1	50.7	125	133	7	46
*C. sakazakii* NCIMB 8272, NCTC 8155	pCS2	NZ_CP012255.1	CP012255	117.8	57.2	103	107	4	46
*C. sakazakii* NCIMB 8272, NCTC 8155	pCS3	NZ_CP012256.1	CP012256	53.4	49.3	55	59	4	46
*C. sakazakii* ATCC BAA-894	pESA2	NC_009779.1	CP000784	31.2	51.6	36	38	2	28
*C. sakazakii* ATCC BAA-894	pESA3	NC_009780.1	CP000785	131.2	56.9	118	120	2	28
*C. sakazakii* SP291	pSP291-2	NC_020261.1	CP004093	52.1	49.2	47	58	11	130
*C. sakazakii* SP291	pSP291-3	NC_020262.1	CP004094	4.4	54.0	4	6	2	130
*C. sakazakii* SP291	pSP291-1	NC_020263.1	CP004092	118.1	57.2	101	108	7	130
*C. turicensis* z3032	pCTU1	NC_013283.1	FN543094	138.3	56.1	109	119	10	45
*C. turicensis* z3032	pCTU2	NC_013284.1	FN543095	22.5	49.2	27	28	1	45
*C. turicensis* z3032	pCTU3	NC_013285.1	FN543096	53.8	50.0	55	62	7	45
*C. dublinensis* LMG 23823	pCDU1	NZ_CP012267.1	CP012267	197.3	56.8	159	163	4	46
*C. condimenti* 1330	pCCO1	NZ_CP012265.1	CP012265	151.5	54.0	126	138	12	46
*C. malonaticus* CMCC 45402	p1	NC_023024.1	CP006732	126.5	57.3	102	104	2	131
*C. malonaticus* CMCC 45402	p2	NC_023025.1	CP006733	55.9	50.4	63	64	1	131
*C. malonaticus* LMG 23826	pCMA1	NZ_CP013941.1	CP013941	126.5	57.3	104	105	1	46
*C. malonaticus* LMG 23826	pCMA2	NZ_CP013942.1	CP013942	52.6	50.3	58	61	3	46

^a^ Information was obtained from NCBI and then summarized. ^b^ The authors would like to remind the readers that many of the plasmids and their genetic loci described in this section possess great nucleotide sequence homology among the different plasmids types (e.g., the virulence plasmids), but the prevalence and distribution of the genetic loci are based on PCR detection using pESA3-pCTU1 sequence inferences.

**Table 4 microorganisms-08-00229-t004:** Prevalence and distribution of virulence factors harbored on the pESA3/pCTU1/pSP291-like incFIB virulence plasmid observed in 570 *Cronobacter* isolates.

Species	No. of Isolates	pESA3/ pCTU1 (incFIB) ^b^	No. of Isolates with the Indicated Plasmidotype (%) ^a^							
			T6SS					FHA	Iron Acquisition	
			*cpa*	Int L	*vgrG*	R end	Int R	*fhaB*	*eitA*	*iucC*
*C. sakazakii*	507	493 (97)	479 (97)	484 (98)	273 (55)	297 (60)	183 (37)	59 (12)	489 (99)	487 (99)
*C. malonaticus*	30	30 (100)	0 (0)	3 (10)	1 (3)	0 (0)	0 (0)	30 (100)	30 (100)	29 (97)
*C. turicensis*	13	13 (100)	0 (0)	1 (8)	2 (15)	0 (0)	0 (0)	13 (100)	13 (100)	13 (100)
*C. muytjensii*	12	9 (75)	0 (0)	0 (0)	0 (0)	0 (0)	0 (0)	0 (0)	9 (100)	1 (11)
*C. dublinensis*	5	4 (80)	0 (0)	0 (0)	0 (0)	0 (0)	0 (0)	0 (0)	4 (100)	1 (25)
*C. universalis*	2	2 (100)	2 (100)	0 (0)	0 (0)	0 (0)	0 (0)	2 (100)	2 (100)	2 (100)
*C. condimenti*	1	1 (100)	0 (0)	0 (0)	0 (0)	0 (0)	0 (0)	0 (0)	1 (100)	1 (100)
Total	570	552 (97)	481 (87)	487 (88)	276 (50)	297 (54)	183 (33)	104 (19)	548 (99)	534 (97)

^a^ The information on strains summarized in this table come from studies reported by Franco et al. [47], Gopinath et al. [68], Jang et al. [129,130], and Tall et al. [147]. Numbers within parentheses are the percentage PCR-positive for each plasmid gene locus in relation to the total number of plasmid (incFIB)-harboring strains of each of the seven *Cronobacter* species as described by Franco et al. [47]. ^b^ The prevalence percentage of pESA3/pCTU1 (incFIB) plasmid among the *Cronobacter* strains was calculated using the total number of strains tested.

**Table 5 microorganisms-08-00229-t005:** Prevalence and distribution of type 1, Beta, Sigma, Pap, and Curli fimbriae gene clusters possessed by the seven *Cronobacter* species ^a^.

*Cronobacter* Species and Species-Associated Fimbriae Types Analyzed Using the PATRIC Database (Number of Strains) [152]							
Fimbriae type	*C. sakazakii* (145)	*C. malonaticus* (2)	*C. turicensis* (2)	*C. muytjensii* (3)	*C. dublinensis* (2)	*C. universalis* (1)	*C. condimenti* (1)
Beta ^b^	136	0	0	0	0	0	0
Sigma ^c^	137	2	1	0	2	1	1
Type1 ^d^	137	2	2	3	2	1	1
Pap ^e^	137	2	2	3	2	1	1
Curli ^f^	0	2	1	0	2	1	1

^a^ This table was adapted from Jang et al. [88]. ^b^ The number of strains for Beta fimbriae reflects the presence of Beta-fimbriae probable major subunit. ^c^ The number of strains with Sigma fimbriae reflects the presence of Sigma-fimbriae chaperone protein, Sigma-fimbriae tip adhesin, Sigma-fimbriae uncharacterized paralogous subunit, and Sigma-fimbriae usher protein. ^d^ The number of strains with Type1 fimbriae reflects the presence of Type 1 fimbriae anchoring protein FimD and Type 1 fimbriae adaptor subunit FimG. ^e^ The number of strains with Pap fimbriae reflects the presence of PapA, P pilus assembly protein (COG3121), and chaperone PapD. ^f^ The number of strains with Curli fimbriae reflect the presence of CsgA or CsgB.

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
