# Peer review of "The Secretion of Toxins and Other Exoproteins of Cronobacter: Role in Virulence, Adaption, and Persistence"

_microorganisms, 2020, doi:10.3390/microorganisms8020229_

Round 1

Reviewer 1 Report

This paper is well written and particularly well-edited and provides an informative update on virulence factors in the form of excreted secondary metabolites by pathogenic species of Cronobacter that have public health and epidemiological significance. The authors, and particularly corresponding author, have an array of recent and relevant publications associated with Cronobacter sakazakii and other pathogenic species of Cronobacter and the review article has new and updated references discussed including a 2019 publication of Microorganisms. I would recommend the paper for publication after undergoing certain minor revisions.

-Am attaching the similarity index report (excluding the references). I think the current similarity index condition of this paper is very minor and perhaps unavoidable while constructing a review paper of this magnitude with over 150 references. However, parts of the manuscript could be further edited using the similarity index report to assure the document meets the very high standards of the Microorganisms. A special emphasis for re-editing of these line numbers are suggested: Lines 40 and 41; Lines 61 to 66; Lines 196 to 200; Lines 521 to 527; Lines 635 to 644; Lines 639 to 702; Lines 782 to 788.

-The CDC and ECDC weblinks provided in pages 856 and 858, respectively, could be incorporated in the references, also references 4, 61, and 154 that currently possess a weblinks could be revised structurally to be in harmony with authors instruction having the access date information. The same goes for line 188, the weblink could be incorporated as a reference per instruction in authors’ guidelines.

-Authors are recommended to assure that the headings and sub-headings of the article are in harmony with Microorganisms guidelines.

-Line 728, I think authors meant to write 4.4 kbp

-Line 797,the reviewer request the table legend to be re-edited for consistency to assure all the letters describing information in the table are superscripted (letters d to f).  

-The article does not currently have authors’ contribution section. Considering that article has 15 authors, perhaps their role could be articulated in detail to assure all authors have contribution for construction and editing of the manuscript that qualifies them to be co-authors in the study.

-Current conclusions of the study are very important and well-constructed, however, they place a heavy emphasis on the need for enhanced epidemiological infrastructure and surveillance for the pathogen. The reviewer suggests incorporating additional information about the secondary metabolites secreted by pathogenic Cronobacter and their role in the development of the enteric and systematic diseases in vulnerable population that is the main theme of the publication.

Author Response

Reviewer1 responses

This paper is well written and particularly well-edited and provides an informative update on virulence factors in the form of excreted secondary metabolites by pathogenic species of Cronobacter that have public health and epidemiological significance. The authors, and particularly corresponding author, have an array of recent and relevant publications associated with Cronobacter sakazakii and other pathogenic species of Cronobacter and the review article has new and updated references discussed including a 2019 publication of Microorganisms. I would recommend the paper for publication after undergoing certain minor revisions.

Author’s response: Thank you so much for your kind words and efforts in reviewing the manuscript.

-Am attaching the similarity index report (excluding the references). I think the current similarity index condition of this paper is very minor and perhaps unavoidable while constructing a review paper of this magnitude with over 150 references. However, parts of the manuscript could be further edited using the similarity index report to assure the document meets the very high standards of the Microorganisms. A special emphasis for re-editing of these line numbers are suggested: Lines 40 and 41; Lines 61 to 66; Lines 196 to 200; Lines 521 to 527; Lines 635 to 644; Lines 639 to 702; Lines 782 to 788.

Author’s response: Thank you for you comments. We have struggled mightily in presenting our work and our discussions in concise ways. Though this is not an excuse we have spent so much time in trying to write concise sentences about our work that sometimes we error on repetition. We have tried once again to try not to be repetitious.  Please see Yellow Highlighted sections corresponding to the above lines.

Lines 40 and 41, now occupy lines 39-42 in the revised manuscript.

Lines 61-66, now occupy lines 59-66 in the revised manuscript.

Lines 196 to 200, now occupy lines 195 to 207 in the revised manuscript.

Lines 521 to 527, now occupy lines Section 8., 556 to 207 in the revised manuscript.

Lines 635 to 644 and 639-702 now occupy lines Section 10., 663 to 706 in the revised manuscript.

Lines 782 to 788 now occupy lines Section 12., lines 813 to 821 in the revised manuscript.

-The CDC and ECDC weblinks provided in pages 856 and 858, respectively, could be incorporated in the references, also references 4, 61, and 154 that currently possess a weblinks could be revised structurally to be in harmony with authors instruction having the access date information. The same goes for line 188, the weblink could be incorporated as a reference per instruction in authors’ guidelines.

Author’s response: Thank you for this suggestion. We have incorporated the weblinks according to Microorganisms style.

-Authors are recommended to assure that the headings and sub-headings of the article are in harmony with Microorganisms guidelines.

Author’s response: Thank you for this important style comment. We have now harmonized the headings and subheading according to Microorganisms guidelines.

-Line 728, I think authors meant to write 4.4 kbp

Authors response: Yes, this is what we mean. Thank you for pointing this out to us.

-Line 797,the reviewer request the table legend to be re-edited for consistency to assure all the letters describing information in the table are superscripted (letters d to f).  

Author’s response: Yes, thank you for this comment. We have carried out your suggestions.

-The article does not currently have authors’ contribution section. Considering that article has 15 authors, perhaps their role could be articulated in detail to assure all authors have contribution for construction and editing of the manuscript that qualifies them to be co-authors in the study.

Authors response: Thank you for pointing this out to us. We have incorporated author contributions, in in lines 903-906 of the revised manuscript.

-Current conclusions of the study are very important and well-constructed, however, they place a heavy emphasis on the need for enhanced epidemiological infrastructure and surveillance for the pathogen. The reviewer suggests incorporating additional information about the secondary metabolites secreted by pathogenic Cronobacter and their role in the development of the enteric and systematic diseases in vulnerable population that is the main theme of the publication.

Author’s response: Thank you so much for this comment. Looking at the concluding remarks, we agree with your assessment. Please see lines 898 to 907 in the revised manuscript.

Reviewer 2 Report

This is an interesting review article on Cronobacter toxins and exoproteins, summarizing  the existing knowledge in the field in a very clear, condense and adequate manner. Taking into consideration that Cronobacter species are relatively new foodborne microorganisms  which are still under investigation, the data presented in this review are of significant interest for the readers of the journal and particularly of the  people working in the fields of food microbiology, safety and quality, in both the research and industry sectors. 

Author Response

Reviewer 2. Authors' response

Thank you so much for your comments and all your efforts that you took to review our manuscript. We very much appreciate you kind words and encouragement. 

Reviewer 3 Report

This is an interesting review of proteobacterial pathogenesis in general and of Cronobacter is particular.  It is comprehensive and will be read by those studying this pathogen.  I have no reservations about the scientific quality.  However, there are numerous examples of typographical errors, awkward phrases, "which" instead of "that", I have made dozens of suggestions.  When the authors have considered these suggestions, and made every effort to optimize the language, the paper will warrant publication.  I do need to upload the marked up copy and will do so in a separate email.

Author Response

Reviewer # 3 author responses. the authors would like to thank the reviewer for their many thoughts, considerations and effort in reviewing the paper.

Comment: Always?

Author’s response: Have removed always. See Lines 45-47 in the revised manuscript.

Comment: temperature-abused? intrinsically and extrinsically?

Author’s response: Have removed temperature abused, but have kept intrinsically and extrinsically.

Comment: Is filth a type of fly?

Author’s response: Yes.

Comment: two words.

Authors’ response> Have corrected Even though. Please see line 71.

Comment: is that surprising? I would guess not. Several pathogens downregulate flagella during infection, or don't make flagella, or hide them.

Authors’ response: Yes, we agree.

Comment: is that really the goal of peristalsis? It's so slow. Isn't the real goal to move food along the GI tract?

Authors’ response: We have revised the sentence to reflect your suggestion. Please see lines 88-90 in the revised Ms.

Comment: one word. Besides, "oftentimes" is vague and could be omitted.

Authors’ response: We have removed "Oftentimes" See line 101 in the revised manuscript.

Comment: have biofilms been documented during infection?

Authors’ response: No not for Cronobacter.  We could only find two studies where the biofilm activity was studied, and these investigators used tissue culture cells.

Comment: "microevolution"?

Authors’ response: We have removed this instance of microevolution See Line 119 in the revised manuscript.

Comment: do they actually colonize flies?

Authors’ response: Yes, according to results reported by Pava-Ripoll in Pava-Ripoll, M.; Pearson, R.E.; Miller A.K.; Ziobro, G.C. Prevalence and relative risk of Cronobacter spp., Salmonella spp., and Listeria monocytogenes associated with the body surfaces and guts of individual filth flies. Appl. Environ. Microbiol. 2012, 78, 7891–7902.

Comment: many microbiologists prefer "microbiota"

Authors’ response: We have changed microflora to microbiota. See line 142 in the revised Ms.

Comment: many bacteria are "commensal pathogens", i.e. able to cause disease in some people and to be harmless in others.

Authors’ response: Thank you for this comment. We have incorporated the sentiment in line 145 of the revised manuscript.

Comment: ST" sounds like "serotypes" Many use the name "genomovars"

Authors’ response: We used sequence type because of the Baldwin and Joseph's reports.

Comment: Loci

Authors’ response: Have changed it to locus. We have made this change. See line 162 in the revised manuscript.

Comment: also?

Authors’ response: We have deleted “also”

Comment: is "uptake" a verb? how about "import"?

Authors’ response: We have made this suggestion See line 170 in the revised manuscript.

Comment: many uses of the word "which" in this document could be replaced with "that" Do a "which hunt"!

Authors’ response: We have replaced “which” in lines 181, 384, 769,789, 880, with ‘that.”

Comment: does the oral cavity suggest a respiratory infection?

Authors’ response: We have deleted reference to the oral cavity. Please see line 188 in the revised manuscript.

Comment: I thought "middle east" was not politically correct, too Eurocentric. perhaps "southern Asia?

Authors’ response: We have made this suggestion. Please see line 201 in the revised Ms.

Comment: typo

Authors’ response: Thanks. Have corrected. See line 211 in the revised Ms. Have replaced it with like

Comment: certainly? It seems highly speculative.

Authors’ response: We have revised this sentence. Please see line 275 in the revised manuscript.

Comment: does the sequence suggest membrane localization, similar to Pla?

Authors’ response: This is unknown at this time.

Comment: i thought you just said it wasn't secreted.

Authors’ response: We have deleted mention of secretion. See line 344 of the revised Ms.

Comment: change "allele" to gene. Allele is the wrong word.

Authors’ response: We have made this suggestion. Please see line 365.

Comment: how does one gene encode two different proteins?

Authors’ response: It is our understanding that the cis-trans activity is due to the proteolytic activity of the enzyme. We have modified this section to read as Cis-trans prolyl isomerases may be related to the mip (macrophage infectivity potentiator) gene [92].

Comment: type 3 injectisomes resemble flagellar basal body; type 4 resemble conjugation systems, only type 6 is phage-like.

Authors’ response: Thank you for the comment. We have incorporated this suggestion. See line 413 in the revised Ms.

Comment: type I what? Did you mean type I pili?

Authors’ response: Yes, we meant fimbriae. We have made this suggestion in line 418 of revised Ms.

Comment: all bacteria

Authors’ response: Have removed reference to Gram-negative.

Comment: never heard of type iv in archaea. Is that described in the reference?

Authors’ response: Yes, T4ss are in Archaea spp. according to Costa et al. [93].

Comment: i'm pretty sure that the ptl genes of B. pertussis release the pertussis toxin into the milieu.

Authors’ response: Yes, it has been found that components of the Bordetella pertussis Ptl transporter, which directs the secretion of the pertussis toxin to mammalian cells, have sequence similarities with the VirB and Tra type IV secretion systems.

Comment: most Ti plasmids have 2 type 4 systems, one for transfer of TDNA into plant cells and another for conjugation of the plasmid. Many conjugation systems closely resemble VirB. Don't over-interpret!

Authors’ response: We have removed reference to A. tumefaciens. Se line 487 in the revised Ms.

Comment: precursor? The C-terminal domain forms a pore for the secretion of the N terminal domain.

Authors’ response:  We have revised this sentence as suggested. Please see line 508-509 in the revised ms.

Comment: milieu? I wasn't aware of this. Most of these systems target other bacteria rather than eukaryotic cells. cooperative? I thought the point was to kill competing bacteria.

Authors’ response: Yes, according to Jani et al. [106] these systems can also turn relationships into commensals.

Comment: typo

Authors’ response: Corrected typo in line 568 of the revised Ms.

Comment why "studies"? Just one study?

Authors’ response: Have corrected this as suggested. please see line 572 in revised Ms.

Comment: regulatory proteins?

Comment: or suggesting that signaling is not critical in those assays.

Authors’ response: We have added this to the sentence. Please see line 583. in revised ms.

Authors’ response: We have made this change in the revised s. Se line 598.

Comment: wildly speculative.

Authors response: Have deleted this sentence in the revised ms. See line 600.

Comments: that? You mean that strains that made AHLs made more exopolysaccharides than those that did not make AHLs?

Authors’ response: Yes, according to Singh et al. Singh, N.; Patil, A.; Prabhune, A.A.; Raghav, M.; Goel, G. Diverse profiles of N–acyl–homoserine lactones in biofilm forming strains of Cronobacter sakazakii. Virulence. 2017, 8, 275–281.

Comments: this sentence needs work.

Authors’ response: We have updated this sentence in the revised Ms. See line 611-613.

Comment: the secretion of a protein is completely unrelated to quorum sensing. I can't see any connection. More generally, QS genes are very often adjacent to the genes they regulate. Is that seen here?

Authors’ response: We have deleted this sentence,

Comment: explained?

Authors’ response: Have changed this to fully understood. See line 619 in revised Ms.

Comment: outwardly? do you mean divergently?

Authors response: we have deleted outwardly.

Comment: comments about GroEL? It seems extremely surprising. Given that GroE is extremely abundant in cytoplasms, I get the OMV prep was contaminated with cells.

Authors’ response: This is the result reported by Kothary et al.

Comments: how would PCR be useful in studying OMV? It is related to sequence homology.

Authors’ response: The interpretation is based on two lines of evidence of being highly conserved among the seven species is based on both PCR analysis for the genes and microarray analysis.

Comment: all one paragraph?

Authors’ response: We have connected the paragraph.

Comment: N-terminal what?

Authors response: We have removed the N-terminal reference.

Comment: what secondary response?

Authors response: Have revised this section extensively. See lines 663 to 706.

Comment: how hot? Do these genes enable growth at high temp or merely ability to withstand the lethal effects of high temp?

Authors’ response: This question is still unanswered,

Comment: any idea why?

Authors’ response: No.

Comment: odd choice of words.

Authors’ response: Have changed this to similar.

Comment: too teleological. How can we ever really know the goal?

Authors’ response: Though speculative, this is the prevailing thought.

Comment: who

Authors’ response: have changed it to who. See line 782 in revised Ms.

Comment: nuclear?

Authors’ response: This was supposed to be nucleator. Have made this change.

Comment: i'm lost. The seven species are current or not?

Authors’ response: there are currently seven species.

Comment: taxonomic opinion is awkward.

Authors response: have changed to scheme. Please see line 873 in the revised manuscript.

Comment: defined

Authors’ response: Have changed to defined . Please see line 884 in the revised ms.

Comment: add "to"

Authors response: Have added to to line.

Comment: typo

Authors response: Have corrected spelling of Officials.

Round 2

Reviewer 3 Report

The revised manuscript is ready to publish!